# One Prompt Fits All: Universal Graph Adaptation for Pretrained Models

**Yongqi Huang**[1][†], **Jitao Zhao**[1][†], **Dongxiao He**[1][*], **Xiaobao Wang**[1], **Yawen Li**[2],
**Yuxiao Huang**[3], **Di Jin**[1], **Zhiyong Feng**[1]

[1]College of Intelligence and Computing, Tianjin University,
[2]School of Economics and Management, Beijing University of Posts and Telecommunications,
[3]Department of Data Science, George Washington University,
[1]{yqhuang, zjtao, hedongxiao, wangxiaobao, jindi, zyfeng}@tju.edu.cn
[2]warmly0716@126.com, [3]yuxiaohuang@gwu.edu

## Abstract

Graph Prompt Learning (GPL) has emerged as a promising paradigm that bridges graph pretraining models and downstream scenarios, mitigating label dependency and the misalignment between upstream pretraining and downstream tasks. Although existing GPL studies explore various prompt strategies, their effectiveness and underlying principles remain unclear. We identify two critical limitations: (1) Lack of consensus on underlying mechanisms: Despite current GPLs have advanced the field, there is no consensus on how prompts interact with pretrained models, as different strategies intervene at varying spaces within the model, i.e., input-level, layer-wise, and representation-level prompts. (2) Limited scenario adaptability: Most methods fail to generalize across diverse downstream scenarios, especially under data distribution shifts (e.g., homophilic-to-heterophilic graphs). To address these issues, we theoretically analyze existing GPL approaches and reveal that representation-level prompts essentially function as fine-tuning a simple downstream classifier, proposing that graph prompt learning should focus on unleashing the capability of pretrained models, and the classifier should adapt to downstream scenarios. Based on our findings, we propose `UniPrompt`, a novel GPL method that adapts any pretrained models, unleashing the capability of pretrained models while preserving the input graph. Extensive experiments demonstrate that our method can effectively integrate with various pretrained models and achieve strong performance across in-domain and cross-domain scenarios.

## 1 Introduction

Graph Prompt Learning (GPL) [1, 2], which aims to design diverse graph prompt strategies, has emerged as a promising and effective alternative paradigm that bridges between graph pretraining [3, 4] and downstream scenarios [5], overcoming the limitations of label dependency and the misalignment between upstream pretraining and downstream tasks [6]. Most GPLs freeze the parameters of the pretrained model and tune specific prompt module. Due to its compatibility with various types of graphs, such as general graphs [6, 7], Knowledge Graphs (KGs) [8, 9, 10] and Text-Attribute Graphs (TAGs) [11, 12], GPL demonstrates strong universality and transferability, thereby improving the fields of few/zero-shot graph learning [12, 13], unified task learning [14, 15, 16], cross-domain graph learning [17, 18], and promoting the development of Graph Foundation Models (GFMs) [19, 20, 21].

---

[†]Equal contribution.
[*]Corresponding author.

39th Conference on Neural Information Processing Systems (NeurIPS 2025).

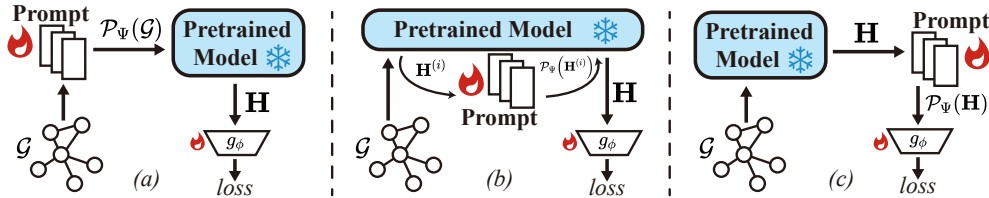

Figure 1: Three different graph prompting mechanisms: input-level prompt (left), layer-wise prompt (middle), and representation-level prompt (right).

Most GPLs can be divided into three categories according to how the prompts are integrated into the pretrained models, as illustrated in Figure 1. Input-level GPLs (Figure 1a), like feature prompt [7] and edge prompt [22, 23], insert prompt modules or soft prompt before the pretrained models, effectively modifying the input graph to align with upstream distributions in pretraining. Representation-level prompts (Figure 1c), including task tokens [6] and prototypical subgraphs [24, 25], applies prompts to the representations generated by the pretrained models, formulating downstream tasks that align with the pretrain objectives. Layer-wise prompt (Figure 1b) [15, 26] combines the prompt with each layer inside the pretrained model, learning the distribution and propagation patterns in each layer. In addition, some works [26, 27] also explore hybrid strategies that place various prompts across different layers or components (input-, representation-level) of the model.

Despite the success of these explorations, our observations indicate that GPLs often experience performance instability or even negative optimization, which also be reported in recent studies [2]. Moreover, the existing prompt methods are complex and diverse. There remains a lack of clear understanding of why graph prompt learning works. Through our analysis of existing methods, we identify two major issues: **1)**. **Lack of consensus on underlying mechanisms**: Although various graph prompt strategies have advanced the field, there remains rare unified understanding of how these prompts interact with pretrained models. As shown in Figure 1, diverse mechanisms such as input-level, representation-level, and layer-wise prompts achieve promising performance. However, they influence the models in different ways, and the underlying interaction mechanisms are still unclear. **2)**. **Limited scenario adaptability**: Most GPLs struggle to achieve good performance on different pretrained models even in the in-domain setting. As shown in Figure 2, only fine-tuning a classifier can achieve or exceed the performance of existing GPLs. In addition, these methods have difficulty achieving excellent performance in a variety of downstream scenarios, especially when the data domains of upstream and downstream scenarios are different (e.g., from a homophilic pretraining graph to a heterophilic downstream graph). To summarize: From the prompting mechanism to the downstream scenario, existing graph prompt learning methods exhibit an adaptation gap.

To investigate the underlying mechanisms of GPL, we conduct a motivation experiment and find that existing representation-level prompt GPLs fail to consistently adapt well to different pretrained models. Moreover, they show no significant performance improvement compared to linear probe (only fine-tune a classifier), which achieves good and stable results. This motivates us to explore the relationship between different types of prompts and linear probe. Through theoretical analysis and discussions, we demonstrate that the representation-level prompt is essentially equivalent to linear probe. This primarily serves to adapt the pretrained model to downstream tasks, which focuses on fitting the outputs of the pretrained models to the downstream labels, struggling to leverage the unique benefits of prompts. As for layer-wise methods, their reliance on layer-wise representations of the pretrained model, combined with their design complexity, makes them unsuitable. In contrast, input-level prompts avoid the limitations and preserve the advantages of prompting, they are the promising among the three categories. Therefore, we propose a perspective: *graph prompt learning should focus on unleashing the capability of pretrained models, and the classifier adapts to downstream scenarios.*

Based on our perspective, we propose `UniPrompt`, a novel GPL method that adapts any pretrained models, leveraging prompt graph while preserving the original structure to unleash the capability of pretrained models. Specifically, we construct a $k$NN graph as the initial prompt graph and adaptively optimize edge weights to guide message passing across nodes. To preserve the input graph, we introduce a bootstrapping strategy that integrates the prompt graph into the original graph topology, preventing model collapse and overfitting. Our main contributions can be summarized as follows:

1. We identify two key issues in existing GPLs: lack of consensus on underlying mechanisms, and limited scenario adaptability. We propose that graph prompt learning should focus on unleashing the capability of pretrained models, and the classifier adapts to downstream scenarios.

2. We propose `UniPrompt`, a novel universal GPL method that adapts any pretrained models. This method leverages a learnable prompt graph while preserving the original structure to unleash the capability of pretrained models.

3. We conduct extensive experiments on homophilic and heterophilic datasets, evaluating in-domain and cross-domain performance under few-shot settings. Experimental results demonstrate that our method consistently outperforms state-of-the-art GPL baselines.

## 2 Notations and Preliminary

**General Graphs.** Given a graph $\mathcal{G} = (\mathcal{V}, \mathcal{E}, \mathbf{X}, \mathbf{Y})$, where $\mathcal{V} = \{v_1, v_2, \cdots, v_N\}$ is the set of nodes, $N = |\mathcal{V}|$, and $\mathcal{E} \subseteq \mathcal{V} \times \mathcal{V}$ is the set of edges. These nodes are associated with feature matrix $\mathbf{X} \in \mathbb{R}^{N \times F}$, $\mathbf{X}_i \in \mathbb{R}^F$ is the feature of $v_i$. The edges can be represented by adjacency matrix $\mathbf{A} \in \{0, 1\}^{N \times N}$, and $\mathbf{A}_{ij} = 1$ iff $(v_i, v_j) \in \mathcal{E}$. Each node $v_i$ is associated with a label $y_i \in \mathbf{Y}$, where $\mathbf{Y}$ denotes the set of all possible class labels. We use $P(\cdot)$ to denote the probability distribution, primarily for distinguishing concepts rather than performing mathematical derivation.

**Fine-Tuning.** In the pretrain-finetune paradigm, given a pretrained graph encoder $f_\theta$ and a downstream trainable projection head $g_\phi$, both parameters $\theta$ and $\phi$ are jointly optimized on a downstream dataset $\mathcal{D} = \{(\mathbf{A}, \mathbf{X}, \mathbf{Y})\}$. The objective is to maximize the log-likelihood of label predictions, which can be formulated as:

$$\max_{\theta, \phi} \frac{1}{|\mathcal{D}|} \sum_{(\mathbf{A}, \mathbf{X}, \mathbf{Y}) \in \mathcal{D}} \sum_{i=1}^{N} \log P\left(y_i \mid g_\phi\left(f_\theta(\mathbf{A}, \mathbf{X})_i\right)\right), \tag{1}$$

where $f_\theta(\mathbf{A}, \mathbf{X})_i$ denotes the node representation of the node $v_i$. As a special case, when freezing $\theta$ (i.e., restricting $\max_\phi$), this reduces to linear probing where only the projection head $g_\phi$ is adapted.

**Graph Prompt Learning.** Compared to fine-tuning, graph prompt learning keeps the pretrained encoder $f_\theta$ frozen while introducing trainable prompt parameters $\Psi$. The optimization objective for all graph prompt learning methods can be expressed as:

$$\max_{\Psi} \frac{1}{|\mathcal{D}|} \sum_{(\mathbf{A}, \mathbf{X}, \mathbf{Y}) \in \mathcal{D}} \sum_{i=1}^{N} \log P\left(y_i \mid \text{Predict}_\Psi\left(\mathbf{A}, \mathbf{X}, v_i; f_\theta\right)\right), \tag{2}$$

where $\mathcal{D} = \{(\mathbf{A}, \mathbf{X}, \mathbf{Y})\}$ is a downstream dataset, $\Psi$ represents all trainable prompt parameters, $f_\theta$ is the frozen pretrained encoder. $\text{Predict}_\Psi(\cdot)$ is a unified prediction function that takes the input graph $(\mathbf{A}, \mathbf{X})$, node $v_i$, the pretrained encoder $f_\theta$, to predict the label of node $v_i$.

For input-level prompt, $\Psi$ acts on the input $(\mathbf{A}, \mathbf{X})$, transforming it before it enters $f_\theta$. For layer-wise prompt, $\Psi$ is integrated within the layers of $f_\theta$. For representation-level prompt, $\Psi$ operates on the representations from $f_\theta$, often as part of the classifier, directly influencing the downstream tasks.

## 3 Motivation Experiments and Analysis

Although graph prompt learning is theoretically distinguished from fine-tuning by freezing the parameters of the pretrained model to retain pretrained knowledge, this seemingly suggests a clear boundary between "unleashing pretrained knowledge" and "adapting to downstream scenarios", which contradicts the problem raised in our Introduction. However, we refute this claim through an experiment, demonstrating that prompt-based methods fall into a "pseudo-adaptation" trap.

To investigate whether graph prompt learning suffers from adaptation bias, we conduct the following experiment. As illustrated in the Figure 2, we select classic GPL methods, `GPPT` [6] and `GraphPrompt` [24], and compare them with fine-tuning that only optimizes the classifier. We employ widely used pretrained models: `DGI` [28], `GRACE` [29], and `GraphMAE` [30], and we keep all other parameters consistent, and observe the differences in downstream training between prompt learning and fine-tuning. Our experimental results reveal that `GPPT` exhibits significant incompatibility when

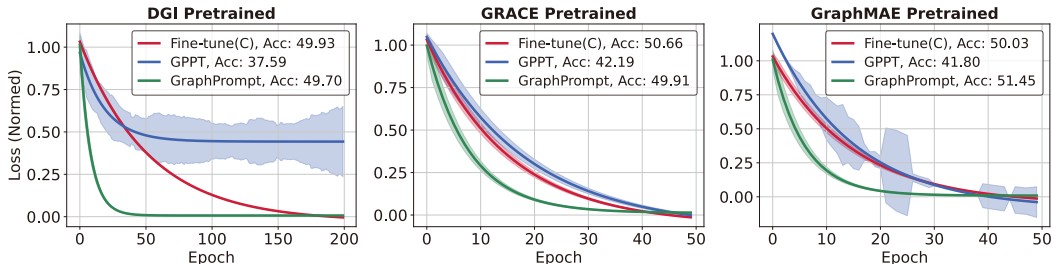

Figure 2: Comparison of fitted normalized loss curves and 1-shot performance across three pretrained models on the *Cora* dataset. **Fine-tune(C)** denotes linear probe, while **GPPT** and **GraphPrompt** are two GPL methods. Shading indicates the standard deviation of the sliding window.

switching pretrained models, whereas `GraphPrompt` maintains stable performance across different pretrained models. Another observation is that even though `GraphPrompt` shows good convergence trends, its performance still falls short of the simple fine-tuning approach in many scenarios, which remains robust across different pretrained models and even outperforms GPL methods in some cases.

This suggests that different graph prompt learning methods heavily depend on the design of the pretrained models. When confronted with varying pretrained models, they often demonstrate incompatibility or suboptimal performance compared to fine-tuning. This raises an important question: Do existing graph prompt learning methods work due to the design of the prompt algorithm, or are they merely benefiting from certain key components in the pretrained model? The answer remains unclear. However, our observations align with prior research [2], when pretrained models are swapped, many GPL methods underperform, while simple fine-tuning can achieve superior results.

Thus, we challenge the current objectives of GPLs: Have existing studies truly succeeded in distinguishing graph prompt learning from fine-tuning? To get deeper insight into the performance differences across various prompt spaces, we focus on a key question: do these differences stem from the prompts' ability to access pretrained knowledge, or from their capacity to adapt to specific tasks? Therefore, we provide a theoretical analysis in this section to understand the underlying mechanisms.

## 4 Mechanism Relationship between Prompts and Classifier

**Definition 4.1** *Given a GNN encoder $\phi(\cdot; \mathcal{G}) : \mathcal{V} \to \mathbb{R}^d$, the representation set $\mathcal{H} = \{h_i \in \mathbb{R}^d \mid v_i \in V\}$ is encoded by $\phi$. The representation of $v_i$ is $h_i = \phi(v_i; \mathcal{G})$. Then, we define the prompt function $T(\cdot)$ and classifier $C(\cdot)$ and baseline classifier $C_0(\cdot)$ as follows:*

$$T(\cdot) : \mathbb{R}^d \to \mathbb{R}^{d'}, \quad C(\cdot) : \mathbb{R}^{d'} \to \mathbb{R}^k, \quad C_0(\cdot) : \mathbb{R}^d \to \mathbb{R}^k, \tag{3}$$

*where $T(\cdot)$ is parameterized as a linear transformation: $T(\mathbf{h}) = \mathbf{W}_T \mathbf{h} + \mathbf{b}_T$, with parameter $\mathbf{W}_T \in \mathbb{R}^{d' \times d}$, $\mathbf{b}_T \in \mathbb{R}^{d'}$. $C(\cdot)$ is implemented as a linear classifier: $C(\mathbf{h}) = \mathbf{W}_C^\top \mathbf{h}$, with $\mathbf{W}_C \in \mathbb{R}^{d' \times k}$. $C_0(\cdot)$ is implemented as a baseline classifier: $C_0(\mathbf{h}) = \mathbf{W}_0^\top \mathbf{h}$, with $\mathbf{W}_0 \in \mathbb{R}^{d \times k}$.*

**Theorem 4.1 (Parameter Objective Equivalence)** *Given a linear prompt function $T(\mathbf{h}) = \mathbf{W}_T \mathbf{h} + \mathbf{b}_T$ and classifier $C(\mathbf{h}) = \mathbf{W}_C^\top \mathbf{h}$, the following properties hold:*

1. ***Function Space Equivalence****: There exists a linear classifier $C'(\mathbf{h}) = \mathbf{W}_{C'}^\top \mathbf{h} + \mathbf{b}_{C'}$ such that $(C \circ T)(\mathbf{h}) = C'(\mathbf{h})$ for all $\mathbf{h}$;*

2. ***Optimization Objective Equivalence****: The optimization problems $\min_{\mathbf{W}_T, \mathbf{b}_T, \mathbf{W}_C} L(C \circ T(\mathbf{h}), y)$ and $\min_{\mathbf{W}_{C'}, \mathbf{b}_{C'}} L(C'(\mathbf{h}), y)$ are equivalent in parameter space and gradient update paths.*

The function space equivalence is guaranteed by Proposition 4.1, and the optimization equivalence is demonstrated in Proposition 4.2.

**Proposition 4.1 (Function Space Equivalence)** *For any linear transformation $T(\mathbf{h}) = \mathbf{W}_T \mathbf{h} + \mathbf{b}_T$ and classifier $C(\mathbf{h}) = \mathbf{W}_C^\top \mathbf{h}$, there exists an equivalent classifier $C'(\mathbf{h}) = \mathbf{W}_{C'}^\top \mathbf{h} + \mathbf{b}_{C'}$ such that $(C \circ T)(\mathbf{h}) = C'(\mathbf{h})$.*

The detailed proof of Proposition 4.1 is provided in Appendix A.1. It shows that a representation-level prompt is functionally equivalent to linear probe $C'$ in the function space. While this equivalence has not been explicitly recognized in prior work. To further clarify the equivalence between a representation-level prompt and linear probe in terms of optimization, we introduce Proposition 4.2.

**Proposition 4.2 (Optimization Objective Equivalence)** *For $(C \circ T)(\mathbf{h})$ and $C'(\mathbf{h})$, we consider the same loss function $L$, the optimization problems $\min_{\mathbf{W}_T, \mathbf{W}_C, \mathbf{b}_T} L((C \circ T)(\mathbf{h}), y)$ and $\min_{\mathbf{W}_{C'}, \mathbf{b}_{C'}} L(C'(\mathbf{h}), y)$ are equivalent in the parameter space.*

The detailed proof of Proposition 4.2 is provided in Appendix A.2. Here we give a brief explanation: Proposition 4.2 demonstrates that the two different optimization formulations, representation-level prompt and linear probe, lead to equivalent parameter updates during optimization. This theoretical equivalence implies that both approaches perform the same underlying optimization. While practical performance differences arise due to the structural complexity of prompts, which may introduce extra challenges, as opposed to linear probe that is typically simpler in design and optimization.

**Discussion 1.** Theorem 4.1 demonstrates that the representation-level prompt is fundamentally equivalent to linear probe, only designing a simple classifier can yield satisfactory results. Therefore, we suggest that graph prompt learning should focus on unleashing the capability of pretrained models, rather than adapting pretrained models to specific downstream scenarios. Specifically, an effective GPL approach should aim to combine the advantages of both mechanisms. Let $\phi_{\text{pre}}(\cdot; \mathcal{G}) = \mathbf{A}\mathbf{X}\mathbf{W}_{\text{pre}}$ be a pretrained GNN encoder with fixed parameters $\mathbf{W}_{\text{pre}}$. We define $\hat{\mathcal{G}} = (\hat{\mathbf{A}}, \hat{\mathbf{X}}) = T(\mathcal{G})$, which modifies the input graph to align the $\phi_{\text{pre}}$. $C(\mathbf{h}) = \mathbf{W}_C^\top \mathbf{h}$ adapts representations to downstream labels. Then, the joint optimization of $(\hat{\mathcal{G}}, \mathbf{W}_C)$ as follows:

$$\min_{\mathbf{W}_C, \hat{\mathcal{G}}} \mathcal{L}_D = - \sum_{v_i \in \mathcal{V}_L} y_i \log \sigma \left( \mathbf{W}_C^\top \hat{\mathbf{A}} \hat{\mathbf{X}} \mathbf{W}_{\text{pre}} \right)_i, \tag{4}$$

where $\mathcal{L}_D$ is the downstream task loss, typically cross-entropy loss. $\hat{\mathbf{A}}\hat{\mathbf{X}}\mathbf{W}_{\text{pre}}$ ensures adaptation with pretrained knowledge, and $\mathbf{W}_C$ minimizes $\mathcal{L}_D$ to adapt downstream task.

**Discussion 2.** Building on Discussion 1, we propose that the input-level and layer-wise prompt mechanisms align with equation 4, where jointly optimizing the prompts and the classifier leads to better performance than using the classifier or prompt only. As for layer-wise methods, their reliance on layer-wise representations of the pretrained model, combined with their design complexity, makes them unsuitable. In contrast, input-level prompts avoid the limitations and preserve the advantages of prompting, which applies prompts to the features or adjacency matrix, helping to reduce structural differences and feature distribution shifts, thereby bridging the gap between upstream pretraining and downstream scenarios. Therefore, we propose that graph prompt learning should focus on unleashing the capability of pretrained models, and the classifier adapts to downstream scenarios. *This viewpoint clearly defines the distinct roles and mechanisms of the two crucial downstream components: the prompt and the classifier.*

# 5 Methodology

In this section, we present our method, `UniPrompt`. Our approach introduces an input-level graph prompt that modifies the graph topology to better align the pretrained models with downstream few-shot tasks. We first introduce an overview of our `UniPrompt`. For a given graph $\mathcal{G} = (\mathbf{A}, \mathbf{X})$ and a frozen pretrained model $f_\theta(\cdot)$, our goal is to adapt it to a downstream task with only a few labeled nodes $\mathcal{V}_L$. Instead of directly fine-tuning $\theta$, `UniPrompt` generates a topological prompt $\tilde{\mathbf{A}}$ with learnable parameters $\Psi$. This prompt is then iteratively fused with the original graph to create a prompt graph, which is fed into the frozen encoder $f_\theta(\cdot)$. Finally, a lightweight classifier $g_\phi(\cdot)$ is trained jointly with the prompt parameters $\Psi$ on the labeled nodes.

**Prompt Initialization.** To enhance prompt adaptability for pretrained models, we consider both in-domain and cross-domain scenarios. In the in-domain case where pretraining and prompt tuning share the same data distribution, the classifier adapts the pretrained model to downstream scenarios while the prompt aligns the pretrained model with downstream inputs. A special case occurs when downstream data is heterophilic, even with matched distributions, the heterophily contradicts the homophily assumption in pretrained models. Existing input-level and layer-wise prompts primarily

process features and tend to overfit in few-shot settings, failing to handle this scenario. In contrast, topological relationships provide more direct and interpretable structural patterns. Therefore, we propose an edge prompt strategy that uses $k$NN to generate a topological prompt with tunable edge weights, formulated as:

$$(\tilde{\mathbf{A}}_{\text{init}})_{ij} = \begin{cases} \mathbf{S}_{ij}, & \text{if } \mathbf{S}_{ij} \in \text{top-k } \{\mathbf{S}_{i\cdot}\}, \\ 0, & \text{otherwise}. \end{cases}, \qquad \mathbf{S}_{ij} = \frac{\mathbf{x}_i \mathbf{x}_j^\top}{\|\mathbf{x}_i\|_2 \|\mathbf{x}_j\|_2}, \tag{5}$$

where $\mathbf{x}_i, \mathbf{x}_j \in \mathbb{R}^F$ are the features for nodes $v_i$ and $v_j$, and $\|\cdot\|_2$ denotes the L2 norm. we select the top-$k$ edges as initial edges and serve as the basis for our learnable prompt.

**Parameterization.** Instead of treating the presence of edges as fixed, we introduce learnable parameters to control the importance of each edge in the initial prompt graph. For every non-zero edge $(\tilde{\mathbf{A}}_{\text{init}})_{ij}$, we associate a learnable scalar weight $w_{ij}$, which forms our set of prompt parameters $\Psi = \{w_{ij}\}$. To enable the model to select the most relevant prompt edges and ensure non-negative weights, we apply a gating mechanism using a scaled and shifted ELU activation function. The prompt adjacency matrix $\tilde{\mathbf{A}}$ is computed as:

$$\tilde{\mathbf{A}}_{ij} = \text{ELU}(w_{ij} \cdot \alpha - \alpha) + 1, \tag{6}$$

where $\alpha$ is a hyperparameter controlling the shape of activation. This parameterization adds learnable edge gating mechanism that can adaptively prune (i.e., approach zero) or amplify topological information for downstream scenarios.

**Bootstrapped Prompt Integration.** After generating the prompt topology $\tilde{\mathbf{A}}$, the challenge lies in its integration with the original adjacency matrix $\mathbf{A}$. While an ideal scenario would involve directly substituting $\mathbf{A}$ with $\tilde{\mathbf{A}}$, this approach proves impractical, particularly in few-shot learning settings, due to risks of severe overfitting and model collapse. Drawing inspiration from Graph Self-Supervised Learning (GSSL) [31, 32] and Graph Structure Learning (GSL) [33], we adopt a bootstrapped integration that iteratively updates the topology rather than directly replacing $\mathbf{A}$. The graph structure fed into the pretrained model at each training epoch is iteratively updated. Let $\hat{\mathbf{A}}^{(t)}$ be the input adjacency matrix at training epoch $t$. The update rule is defined as:

$$\hat{\mathbf{A}}^{(t)} = \tau \hat{\mathbf{A}}^{(t-1)} + (1 - \tau)\tilde{\mathbf{A}}, \tag{7}$$

where $\hat{\mathbf{A}}^{(0)} = \mathbf{A}$ is the original adjacency matrix, and the temperature coefficient $\tau \in [0, 1]$ controls the balance between original and prompt topology.

**Optimization Objective.** For subsequent epochs, the input to the pretrained model becomes $\hat{\mathcal{G}} = (\hat{\mathbf{A}}, \mathbf{X})$. Through UniPrompt, we process $\hat{\mathcal{G}}$ via the pretrained model to obtain node representations $\mathbf{H}$. Our empirical results in Figure 2 demonstrate that linear probe achieves comparable performance to existing GPL methods in few-shot settings. This demonstrates the capability of the classifier to adapt to downstream scenario, confirming its effectiveness in this configuration. Therefore, we incorporate a learnable projection head $g_\phi$ in the representation and jointly optimize it with UniPrompt. The overall framework is optimized via the following equation:

$$\min_{\phi, \Psi} \frac{1}{|\mathcal{V}_L|} \sum_{v_i \in \mathcal{V}_L} \ell_D \left( g_\phi \left( f_\theta \left( p_\Psi(\mathbf{A}, \mathbf{X}) \right)_i \right), y_i \right), \tag{8}$$

where $y_i$ is the ground-truth label of node $v_i \in \mathcal{V}_L$, and $\ell_D$ is the downstream task loss, i.e., the cross-entropy loss for classification tasks.

## 6 Experiments

### 6.1 Experimental Setup

We evaluate the effectiveness of UniPrompt [1] using nine node classification datasets, including three homophilic datasets *Cora* [34], *CiteSeer* [34] and *PubMed* [34], and six heterophilic datasets *Cornell* [35], *Texas* [35], *Wisconsin* [35], *Chameleon* [35], *Actor* [35] and *Squirrel* [35]. For in-domain

---

[1]Code is available at: https://github.com/hedongxiao-tju/UniPrompt

Table 1: In-domain node classification. Accuracy on 1-shot node classification tasks over three pretrained models and nine datasets. The best results in each pretrain strategy are highlighted in **bold**, and the runner-up with an underline.

| Pretrain | Methods | Cora | CiteSeer | PubMed | Cornell | Texas | Wisconsin | Chameleon | Actor | Squirrel |
|---|---|---|---|---|---|---|---|---|---|---|
| DGI | Fine-tune | $\underline{50.22}_{\pm9.28}$ | $42.58_{\pm8.87}$ | $53.90_{\pm8.30}$ | $\underline{35.23}_{\pm8.84}$ | $\underline{37.50}_{\pm13.57}$ | $\underline{33.91}_{\pm10.56}$ | $\mathbf{24.42}_{\pm3.19}$ | $\underline{21.36}_{\pm3.28}$ | $22.27_{\pm4.10}$ |
| | Linear-probe | $49.77_{\pm9.74}$ | $43.16_{\pm7.60}$ | $\underline{55.76}_{\pm9.43}$ | $34.56_{\pm8.60}$ | $36.21_{\pm13.77}$ | $28.71_{\pm9.38}$ | $23.64_{\pm2.17}$ | $21.33_{\pm2.62}$ | $22.82_{\pm4.10}$ |
| | GPPT | $37.59_{\pm7.38}$ | $36.01_{\pm6.33}$ | $51.56_{\pm6.64}$ | $29.01_{\pm8.32}$ | $31.26_{\pm8.51}$ | $28.56_{\pm6.50}$ | $22.15_{\pm2.50}$ | $19.81_{\pm1.63}$ | $20.71_{\pm1.24}$ |
| | GraphPrompt | $49.70_{\pm10.27}$ | $43.98_{\pm7.61}$ | $46.32_{\pm7.80}$ | $22.29_{\pm6.44}$ | $27.62_{\pm11.08}$ | $22.62_{\pm8.14}$ | $23.59_{\pm2.54}$ | $19.84_{\pm2.79}$ | $\underline{22.85}_{\pm3.28}$ |
| | All-in-one | $32.10_{\pm6.50}$ | $28.77_{\pm3.12}$ | $35.87_{\pm7.53}$ | $26.67_{\pm12.42}$ | $31.53_{\pm13.14}$ | $24.82_{\pm8.77}$ | $22.41_{\pm3.58}$ | $19.93_{\pm5.23}$ | $21.61_{\pm5.87}$ |
| | GPF | $\mathbf{51.68}_{\pm9.52}$ | $43.11_{\pm5.76}$ | $53.09_{\pm9.66}$ | $26.76_{\pm8.87}$ | $34.04_{\pm15.54}$ | $26.59_{\pm8.94}$ | $23.29_{\pm3.67}$ | $20.31_{\pm4.17}$ | $21.66_{\pm3.28}$ |
| | GPF+ | $48.66_{\pm6.80}$ | $\underline{44.89}_{\pm6.61}$ | $52.58_{\pm9.79}$ | $25.23_{\pm8.76}$ | $28.55_{\pm13.49}$ | $22.82_{\pm8.89}$ | $22.98_{\pm3.66}$ | $20.81_{\pm3.08}$ | $21.56_{\pm3.68}$ |
| | EdgePrompt | $42.05_{\pm6.36}$ | $38.54_{\pm6.37}$ | $47.67_{\pm4.73}$ | $28.00_{\pm8.51}$ | $31.32_{\pm15.82}$ | $32.64_{\pm11.87}$ | $23.17_{\pm3.78}$ | $\underline{21.36}_{\pm2.76}$ | $21.99_{\pm2.50}$ |
| | EdgePrompt+ | $41.74_{\pm6.73}$ | $36.10_{\pm6.15}$ | $46.73_{\pm5.53}$ | $28.37_{\pm7.94}$ | $33.75_{\pm13.57}$ | $33.38_{\pm11.81}$ | $22.95_{\pm3.78}$ | $20.16_{\pm2.65}$ | $21.74_{\pm2.10}$ |
| | UniPrompt | $49.95_{\pm10.48}$ | $\mathbf{45.57}_{\pm8.63}$ | $\mathbf{57.17}_{\pm7.11}$ | $\mathbf{51.13}_{\pm13.26}$ | $\mathbf{48.21}_{\pm15.95}$ | $\mathbf{58.75}_{\pm13.41}$ | $\underline{23.75}_{\pm4.02}$ | $\mathbf{25.38}_{\pm4.86}$ | $\mathbf{24.20}_{\pm2.35}$ |
| GRACE | Fine-tune | $\underline{48.59}_{\pm9.20}$ | $\underline{46.16}_{\pm6.30}$ | $\mathbf{57.97}_{\pm7.55}$ | $34.18_{\pm10.18}$ | $31.52_{\pm13.08}$ | $32.23_{\pm8.96}$ | $26.22_{\pm2.73}$ | $\underline{20.81}_{\pm2.86}$ | $21.16_{\pm2.57}$ |
| | Linear-probe | $46.22_{\pm7.92}$ | $46.10_{\pm6.32}$ | $\underline{57.87}_{\pm7.60}$ | $\underline{34.92}_{\pm9.74}$ | $\underline{34.84}_{\pm15.65}$ | $31.66_{\pm8.18}$ | $24.27_{\pm3.84}$ | $20.53_{\pm3.11}$ | $20.81_{\pm1.82}$ |
| | GPPT | $42.19_{\pm6.42}$ | $37.42_{\pm9.10}$ | $47.62_{\pm7.86}$ | $27.88_{\pm8.09}$ | $32.97_{\pm13.84}$ | $26.53_{\pm8.72}$ | $25.46_{\pm5.43}$ | $19.20_{\pm4.16}$ | $21.56_{\pm2.30}$ |
| | GraphPrompt | $\mathbf{49.91}_{\pm9.60}$ | $35.64_{\pm8.35}$ | $53.63_{\pm9.01}$ | $23.20_{\pm5.83}$ | $30.19_{\pm13.63}$ | $23.07_{\pm6.73}$ | $\mathbf{28.28}_{\pm4.38}$ | $19.15_{\pm3.39}$ | $\underline{22.48}_{\pm2.66}$ |
| | All-in-one | $34.53_{\pm5.86}$ | $24.06_{\pm6.18}$ | $34.51_{\pm7.45}$ | $22.17_{\pm5.40}$ | $27.37_{\pm13.79}$ | $36.17_{\pm6.32}$ | $19.46_{\pm0.29}$ | $19.04_{\pm4.30}$ | $22.03_{\pm2.46}$ |
| | GPF | $48.41_{\pm8.17}$ | $36.78_{\pm4.96}$ | $50.59_{\pm7.18}$ | $28.21_{\pm8.25}$ | $29.98_{\pm14.44}$ | $27.58_{\pm5.74}$ | $25.25_{\pm4.33}$ | $20.20_{\pm2.65}$ | $20.80_{\pm3.05}$ |
| | GPF+ | $47.06_{\pm8.14}$ | $44.46_{\pm6.76}$ | $51.38_{\pm7.19}$ | $28.91_{\pm8.85}$ | $31.49_{\pm14.92}$ | $27.49_{\pm8.38}$ | $26.03_{\pm4.37}$ | $20.13_{\pm2.90}$ | $21.41_{\pm2.96}$ |
| | EdgePrompt | $41.95_{\pm8.19}$ | $36.65_{\pm6.07}$ | $48.20_{\pm10.08}$ | $31.85_{\pm6.19}$ | $29.27_{\pm11.99}$ | $38.62_{\pm8.25}$ | $23.23_{\pm3.25}$ | $20.78_{\pm2.67}$ | $21.76_{\pm1.66}$ |
| | EdgePrompt+ | $45.32_{\pm9.03}$ | $35.80_{\pm6.37}$ | $50.01_{\pm11.96}$ | $32.13_{\pm7.42}$ | $31.95_{\pm6.51}$ | $\underline{38.68}_{\pm7.73}$ | $23.79_{\pm3.31}$ | $20.63_{\pm2.95}$ | $20.97_{\pm1.06}$ |
| | UniPrompt | $44.73_{\pm10.78}$ | $\mathbf{47.53}_{\pm10.13}$ | $\underline{57.88}_{\pm4.80}$ | $\mathbf{52.80}_{\pm11.08}$ | $\mathbf{45.38}_{\pm19.87}$ | $\mathbf{50.98}_{\pm15.38}$ | $\underline{26.67}_{\pm2.51}$ | $\mathbf{26.23}_{\pm4.46}$ | $\mathbf{23.98}_{\pm2.53}$ |
| GraphMAE | Fine-tune | $45.92_{\pm9.67}$ | $36.47_{\pm8.35}$ | $54.29_{\pm9.52}$ | $\underline{35.82}_{\pm11.30}$ | $37.07_{\pm14.08}$ | $\underline{33.54}_{\pm10.16}$ | $22.08_{\pm3.19}$ | $\underline{20.85}_{\pm1.68}$ | $\underline{21.32}_{\pm2.65}$ |
| | Linear-probe | $\underline{50.13}_{\pm12.06}$ | $\underline{48.08}_{\pm6.96}$ | $\mathbf{58.61}_{\pm8.34}$ | $32.27_{\pm11.28}$ | $\underline{38.32}_{\pm13.61}$ | $28.40_{\pm8.67}$ | $\underline{23.02}_{\pm2.08}$ | $20.56_{\pm2.91}$ | $21.05_{\pm1.87}$ |
| | GPPT | $41.80_{\pm8.72}$ | $31.96_{\pm5.26}$ | $49.10_{\pm8.06}$ | $26.74_{\pm7.86}$ | $35.16_{\pm15.12}$ | $25.86_{\pm8.65}$ | $21.87_{\pm3.25}$ | $19.36_{\pm3.72}$ | $20.59_{\pm1.80}$ |
| | GraphPrompt | $\mathbf{51.45}_{\pm9.63}$ | $37.07_{\pm6.19}$ | $50.87_{\pm6.84}$ | $23.82_{\pm7.50}$ | $26.04_{\pm11.72}$ | $26.78_{\pm9.77}$ | $22.05_{\pm2.61}$ | $17.82_{\pm2.84}$ | $20.71_{\pm4.21}$ |
| | All-in-one | $28.96_{\pm4.87}$ | $31.72_{\pm2.78}$ | $39.99_{\pm6.21}$ | $22.33_{\pm6.43}$ | $29.71_{\pm20.15}$ | $29.85_{\pm13.99}$ | $20.13_{\pm1.81}$ | $21.08_{\pm2.17}$ | $20.39_{\pm0.93}$ |
| | GPF | $46.74_{\pm8.50}$ | $40.07_{\pm8.34}$ | $55.38_{\pm7.53}$ | $27.21_{\pm7.70}$ | $28.98_{\pm14.02}$ | $25.65_{\pm8.15}$ | $22.30_{\pm2.58}$ | $20.20_{\pm3.78}$ | $20.19_{\pm0.80}$ |
| | GPF+ | $43.30_{\pm11.40}$ | $40.15_{\pm6.79}$ | $52.92_{\pm7.95}$ | $26.38_{\pm8.48}$ | $34.83_{\pm16.64}$ | $26.79_{\pm9.14}$ | $22.35_{\pm3.60}$ | $20.44_{\pm3.64}$ | $20.26_{\pm0.57}$ |
| | EdgePrompt | $39.16_{\pm9.95}$ | $35.03_{\pm6.90}$ | $49.79_{\pm7.47}$ | $25.26_{\pm7.20}$ | $35.02_{\pm16.61}$ | $26.02_{\pm8.60}$ | $22.27_{\pm3.90}$ | $19.93_{\pm3.19}$ | $20.16_{\pm1.09}$ |
| | EdgePrompt+ | $40.11_{\pm10.12}$ | $37.13_{\pm6.93}$ | $50.77_{\pm7.91}$ | $26.15_{\pm7.77}$ | $34.21_{\pm15.55}$ | $25.84_{\pm9.35}$ | $22.47_{\pm3.82}$ | $20.20_{\pm3.00}$ | $20.73_{\pm1.10}$ |
| | UniPrompt | $47.05_{\pm9.17}$ | $\mathbf{49.29}_{\pm11.20}$ | $\underline{57.47}_{\pm6.86}$ | $\mathbf{51.28}_{\pm12.45}$ | $\mathbf{49.83}_{\pm17.85}$ | $\mathbf{61.38}_{\pm13.58}$ | $\mathbf{24.29}_{\pm3.64}$ | $\mathbf{23.35}_{\pm3.57}$ | $\mathbf{22.08}_{\pm2.03}$ |

settings, we use DGI [28], GRACE [29], and GraphMAE [30] as pretrained models, and we compare our method with two baseline tuning methods, and seven classic and state-of-the-art GPL methods, including Fine-tune, Linear-probe (fine-tune classifier only), GPPT [6], GraphPrompt [24], All-in-one [14], GPF [7], GPF-plus [7], EdgePrompt [23], and EdgePrompt-plus [23]. For cross-domain settings, we adopt FUG [36] as the pretrained model, and we compare our method with four types of baseline methods, including: (1) Semi-Supervised baselines: GCN [3], GAT [4]. (2) Graph Self-Supervised Learning baselines: DGI [28], GraphCL [37]. (3) Graph Prompt Learning baselines: GPPT [6], GPF [7]. (4) Multi-domain Graph Pre-train baselines: GCOPE [38], MDGPT [39], MDGFM [27]. In our experiments. To ensure performance reliability, we perform 20 repeated runs for each of 5 fixed random seeds, reporting averaged results over 100 trials. Detailed information about the experimental setup can be found in the Appendix B.1.

## 6.2 In-Domain Node Classification

**1-shot node classification on different pretrained models.** We report 1-shot node classification on nine datasets using three pretrained models. As shown in Table 1, our method outperforms existing GPLs across most datasets under different pretrained models. Specifically, we observe the most significant improvements on the *Cornell*, *Texas*, and *Wisconsin* datasets, where our method surpasses all existing GPLs. This is primarily because these GPL baselines struggle to adapt downstream datasets to the pretrained model, particularly for heterophilic graphs, which pose a significant challenge. In larger heterophilic datasets like *Actor* and *Squirrel*, the dense connectivity and size of these datasets make baselines challenging in the 1-shot setting. Existing methods are unable to leverage representation-level prompts or directly process features or edges to improve performance. As a result, these methods suffer from model collapse or overfitting. For homophilic datasets, such as *Cora* and *CiteSeer*, all baselines perform well, resulting in limited improvement for our approach. However, for *PubMed*, which has fewer classes, our prompt graph introduces additional homophilic edges, providing an advantage over other GPL baselines. Moreover, we observe that the choice of pretrained model has an impact on downstream prompt tuning. For example, on the *Chameleon* dataset, under the DGI and GRACE pretrained settings, all baselines perform comparably to or even better than our method. However, when switching to GraphMAE, the performance of all methods drops sharply. Similar trends are observed on *CiteSeer* and *PubMed*, where our model demonstrates greater stability compared to other baselines.

**1/3/5-shot Node Classification Performance on DGI.** To further demonstrate the adaptability of our method, we conduct 3-shot and 5-shot experiments on GPL baselines using the DGI-pretrained

Table 2: Cross-domain node classification. Accuracy on 1-shot node classification tasks over six datasets. Each column represents a test domain, while others are train domains. The best results are highlighted in **bold**, and the runner-up with an underline. Methods with ∗ are reported from [27].

| Methods | Cora | Citeseer | PubMed | Cornell | Squirrel | Chameleon |
|---------|------|----------|--------|---------|----------|-----------|
| GCN* | $28.57_{\pm 5.07}$ | $31.27_{\pm 4.53}$ | $40.55_{\pm 5.65}$ | $31.81_{\pm 4.71}$ | $20.00_{\pm 0.29}$ | $24.17_{\pm 5.21}$ |
| GAT* | $28.40_{\pm 6.25}$ | $30.76_{\pm 5.40}$ | $39.99_{\pm 4.96}$ | $28.03_{\pm 13.19}$ | $21.55_{\pm 2.30}$ | $23.93_{\pm 4.11}$ |
| DGI* | $29.30_{\pm 5.82}$ | $30.03_{\pm 4.88}$ | $41.85_{\pm 7.78}$ | $31.54_{\pm 15.66}$ | $21.15_{\pm 1.68}$ | $21.73_{\pm 5.47}$ |
| GraphCL* | $34.94_{\pm 6.49}$ | $30.58_{\pm 4.58}$ | $40.37_{\pm 7.81}$ | $27.15_{\pm 12.64}$ | $21.42_{\pm 2.23}$ | $22.49_{\pm 3.02}$ |
| GPPT* | $17.52_{\pm 5.52}$ | $21.45_{\pm 3.45}$ | $36.56_{\pm 5.31}$ | $25.09_{\pm 2.92}$ | $20.09_{\pm 0.91}$ | $24.53_{\pm 2.55}$ |
| GPF* | $37.84_{\pm 11.07}$ | $37.61_{\pm 8.87}$ | $46.36_{\pm 7.48}$ | $34.54_{\pm 7.73}$ | $21.92_{\pm 3.50}$ | $\underline{25.90}_{\pm 8.51}$ |
| GCOPE* | $34.23_{\pm 8.16}$ | $39.05_{\pm 8.82}$ | $44.85_{\pm 6.72}$ | $34.02_{\pm 11.94}$ | $22.46_{\pm 1.96}$ | $24.61_{\pm 3.99}$ |
| MDGPT* | $39.54_{\pm 9.02}$ | $39.24_{\pm 8.95}$ | $45.39_{\pm 11.01}$ | $33.58_{\pm 10.38}$ | $22.35_{\pm 3.77}$ | $23.68_{\pm 1.56}$ |
| MDGFM* | $\underline{44.83}_{\pm 7.41}$ | $\underline{42.18}_{\pm 6.41}$ | $\underline{46.84}_{\pm 7.31}$ | $\underline{40.77}_{\pm 5.96}$ | $\underline{24.30}_{\pm 3.26}$ | $\mathbf{28.36}_{\pm 3.65}$ |
| UniPrompt(Ours) | $\mathbf{45.37}_{\pm 9.08}$ | $\mathbf{43.25}_{\pm 9.61}$ | $\mathbf{55.01}_{\pm 3.36}$ | $\mathbf{51.58}_{\pm 9.91}$ | $\mathbf{25.29}_{\pm 3.86}$ | $25.14_{\pm 5.65}$ |

model. As shown in Figure 3, our method consistently outperforms existing GPL baselines across most heterophilic datasets. Performance improvements are observed on *Cornell*, *Texas*, *Wisconsin*, and *Actor*, indicating that our approach avoids overfitting and makes use of the label information. On *CiteSeer* and *PubMed*, our method also outperforms existing GPL baselines, demonstrating its effectiveness when the dataset matches the homophily assumption of pretrained model. On *Cora* and *Chameleon*, the advantages of our method become more pronounced as more labels are introduced, gradually surpassing current GPL baselines. Similar experiments are conducted on GRACE and GraphMAE, further validating the generalization capability of our method. Detailed results and analysis can be found in the Appendix B.4.

## 6.3 Cross-Domain Node Classification

To further demonstrate the broad scenario adaptability of our method compared to existing approaches, we conduct experiments under both 3/5-shot and challenging multi-domain pretraining settings. In these settings, not only are the upstream and downstream datasets entirely different, but the multiple source domains within the pretraining also differ in both structure and semantics.

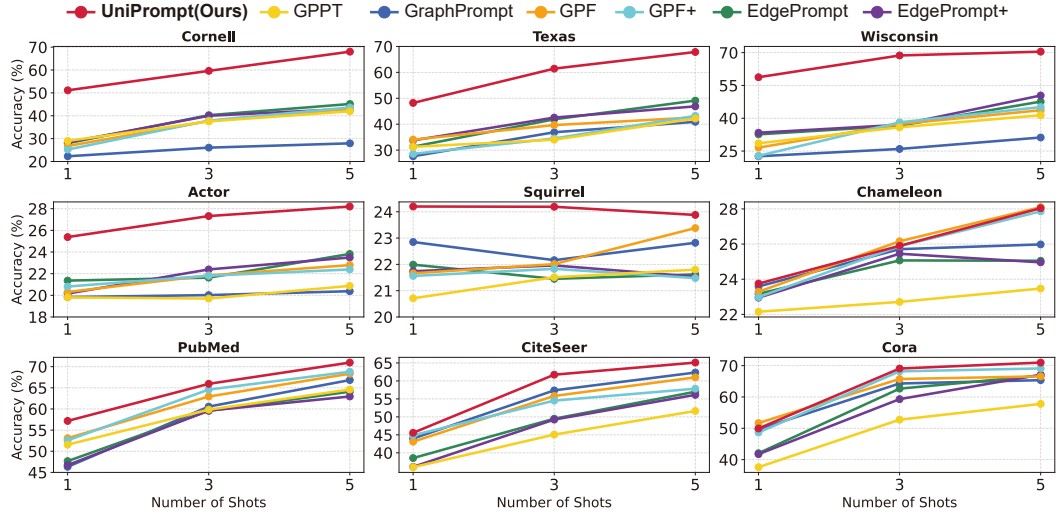

Figure 3: In-domain node classification experiments over nine datasets under different shot settings using DGI as the pretrained model.

**1-shot Cross-Domain Node Classification.** As shown in Table 2, our method outperforms various existing baselines, and competes with or surpasses state-of-the-art GFMs. Our approach achieves improvement on the *PubMed* and *Cornell* datasets. This improvement can be attributed to introduce connections among semantically similar nodes. In scenarios such as *PubMed*, which has few classes, and *Cornell*, which is a sparse graph, our method enables a more effective adaptation of the pretrained model. In contrast, on *Squirrel* and *Chameleon* datasets, where graphs exhibit low homophily and dense inter-class connectivity, the performance differences across methods are less distinct.

Table 3: Cross-domain node classification. Accuracy on 3/5-shot node classification tasks over six datasets. Each column represents a test domain, while others are train domains. The best results are highlighted in **bold**, and the runner-up with an underline. Methods with ∗ are reported from [27].

| Methods | Cora(5) | CiteSeer(5) | Pubmed(5) | Cornell(3) | Squirrel(3) | Chameleon(5) |
|---|---|---|---|---|---|---|
| GCN* | $60.15_{\pm5.33}$ | $45.54_{\pm4.71}$ | $57.82_{\pm8.26}$ | $39.53_{\pm13.57}$ | $21.61_{\pm4.22}$ | $22.09_{\pm0.99}$ |
| GAT* | $59.79_{\pm3.89}$ | $50.48_{\pm2.94}$ | $57.55_{\pm9.37}$ | $34.53_{\pm13.01}$ | $20.11_{\pm3.11}$ | $20.83_{\pm1.52}$ |
| DGI* | $56.76_{\pm11.29}$ | $42.67_{\pm8.98}$ | $54.04_{\pm11.59}$ | $43.22_{\pm5.84}$ | $20.23_{\pm1.12}$ | $27.68_{\pm5.21}$ |
| GraphCL* | $61.59_{\pm5.71}$ | $47.05_{\pm6.85}$ | $58.50_{\pm7.38}$ | $32.77_{\pm6.23}$ | $21.18_{\pm0.96}$ | $27.45_{\pm2.58}$ |
| GPPT* | $43.67_{\pm7.11}$ | $47.31_{\pm6.93}$ | $40.47_{\pm10.17}$ | $34.69_{\pm8.54}$ | $22.14_{\pm1.53}$ | $28.25_{\pm1.39}$ |
| GPF* | $51.21_{\pm11.44}$ | $56.90_{\pm8.84}$ | $58.76_{\pm7.70}$ | $38.17_{\pm8.15}$ | $21.62_{\pm3.10}$ | $28.09_{\pm4.93}$ |
| GCOPE* | $54.63_{\pm3.98}$ | $53.18_{\pm4.47}$ | $57.74_{\pm2.73}$ | $48.21_{\pm11.97}$ | $21.37_{\pm4.20}$ | $25.50_{\pm1.23}$ |
| MDGPT* | $59.64_{\pm5.73}$ | $52.71_{\pm5.71}$ | $58.65_{\pm7.54}$ | $35.18_{\pm8.90}$ | $21.42_{\pm4.16}$ | $26.18_{\pm5.18}$ |
| MDGFM* | $\underline{64.56}_{\pm7.29}$ | $\mathbf{61.24}_{\pm4.82}$ | $\underline{63.50}_{\pm5.81}$ | $\underline{49.56}_{\pm6.92}$ | $\underline{23.00}_{\pm4.39}$ | $\underline{30.54}_{\pm2.87}$ |
| UniPrompt(Ours) | $\mathbf{65.64}_{\pm3.53}$ | $\underline{59.37}_{\pm2.78}$ | $\mathbf{65.09}_{\pm2.51}$ | $\mathbf{52.09}_{\pm5.37}$ | $\mathbf{26.70}_{\pm3.78}$ | $\mathbf{31.38}_{\pm2.67}$ |

**3/5-shot Cross-Domain Node Classification.** As shown in Table 3, the availability of additional labels significantly boosts performance compared to the 1-shot scenario. On *Squirrel* and *Chameleon*, our method achieves notable improvements and outperforms existing approaches. Moreover, our method maintains superior performance on sparse graphs like *Cornell* and few-class datasets like *PubMed*. In contrast, on *Cora* and *CiteSeer*, which exhibit high homophily, most baselines, including supervised baselines and GFMs, perform well when more labeled data are available. The performance gap between our method and existing GPL baselines becomes less pronounced in these two datasets. Overall, our method still maintains advantage, particularly as a GPL approach, which demonstrates the effectiveness in adapting to pretrained models.

## 6.4  Hyperparameter Analysis

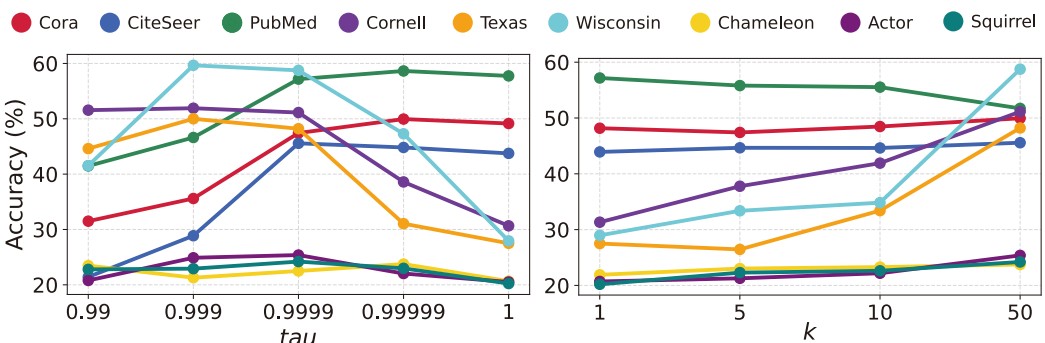

Figure 4: Hyperparameter analysis of $\tau$ and $k$ for 1-shot node classification with DGI pretrained.

We conduct hyperparameter analysis on $\tau$ and $k$ under DGI pretrained and 1-shot settings. As shown in Figure 4, for $\tau$, a notable observation is that heterophilic graphs require lower $\tau$ values (typically converging to 0.999 or 0.9999) to achieve performance gains, which validates the necessity of the prompt graph. For homophilic graphs, performance stabilizes when $\tau \geq 0.9999$, consistent with existing research findings [35] that these graphs align with the homophily assumption of the pretrained models. For smaller $\tau$ values (e.g., $\tau$=0.99), only *Cornell* maintains performance while other datasets

degrade, highlighting the necessity of input graph and demonstrating the robustness against model collapse. When $\tau$=1.0, no prompt graph is injected, corresponding to the ablation of our method. For $k$, The performance gains are most pronounced on sparse heterophilic graphs, i.e., *Cornell*, *Texas*, and *Wisconsin*. For larger datasets *Chameleon*, *Actor*, and *Squirrel*, the prompt graph provides similarity information that improves performance. For *Cora* and *CiteSeer*, performance remains stable across $k$ values. However, *PubMed* performance drops when $k$ reaches 50, which attributes to the effects of high homophily and limited classes.

## 7    Conclusion

In this work, we categorize existing Graph Prompt Learning (GPL) methods based on their mechanisms and conduct an analysis of them. Through this analysis, we identify a key problem in existing GPL methods: the adaptation gap between upstream pretraining and downstream scenarios. We decompose this issue into two aspects: lack of consensus on underlying mechanisms, and limited scenario adaptability. Through motivation experiments and theoretical analysis, we reveal that the representation-level prompt is fundamentally equivalent to fine-tuning a simple downstream classifier. This primarily serves to adapt the pretrained model to downstream tasks, rather than unleashing its inherent capabilities. We propose a perspective: graph prompt learning should focus on unleashing the capability of pretrained models, and the classifier adapts to downstream scenarios. Based on our perspective, we propose `UniPrompt`, a novel GPL method that adapts any pretrained models, which leverages prompt-generated topology while preserving the original structure to unleash the capability of pretrained models. We evaluate `UniPrompt` on a comprehensive set of datasets, including homophilic and heterophilic graphs, under few-shot learning settings. The results demonstrate that `UniPrompt` consistently outperforms state-of-the-art baselines in both in-domain and cross-domain scenarios. Overall, our work provides new perspectives on the design principles of GPL, improving the fields of few/zero-shot graph learning, unifying downstream graph tasks, cross-domain graph learning, and promoting the development of Graph Foundation Models.

## 8    Acknowledgments

This work was supported by the National Natural Science Foundation of China (No. 62422210, No. 62276187, No. 62302333, No.62372323 and No. 92370111), the National Key Research and Development Program of China (No. 2023YFC3304503), and the Hebei Natural Science Foundation (No. F2024202047).

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

# A  Proofs

## A.1  Proof for Proposition 4.1

*Proof:*  $(C \circ T)(\mathbf{h}) = \mathbf{W}_C^\top(\mathbf{W}_T\mathbf{h} + \mathbf{b}_T) = (\mathbf{W}_T^\top\mathbf{W}_C)^\top\mathbf{h} + \mathbf{W}_C^\top\mathbf{b}_T$. Then, we let $\mathbf{W}_{C'} = \mathbf{W}_T^\top\mathbf{W}_C, \mathbf{b}_{C'} = \mathbf{W}_C^\top\mathbf{b}_T$, we can get $(C \circ T)(\mathbf{h}) = \mathbf{W}_{C'}^\top\mathbf{h} + \mathbf{b}_{C'} \equiv C'(\mathbf{h})$. Therefore, we conclude that any linear prompt combination can be represented as a linear classifier with a bias term.  □

## A.2  Proof for Proposition 4.2

*Proof:*  For any parameters of objective function $\mathbf{W}_{C'}$ and $\mathbf{b}_{C'}$, there exists $\mathbf{W}_T$ and $\mathbf{W}_C$:

$$\mathbf{W}_C = (\mathbf{W}_T^\top)^{-1}\mathbf{W}_{C'}, \quad \mathbf{b}_T = \mathbf{W}_C^\dagger\mathbf{b}_{C'}, \tag{9}$$

where $\mathbf{W}_C^\dagger$ is the pseudo-inverse matrix of $\mathbf{W}_C$. The mapping above is unique when $\mathbf{W}_C^\dagger = \left(\mathbf{W}_C^\top\mathbf{W}_C\right)^{-1}\mathbf{W}_C^\top$ and $\mathbf{W}_C$ has full column rank. We calculate the gradient update paths of the two optimization methods respectively. For the original gradient of $\mathbf{W}_C$ and $\mathbf{W}_T$, we have the following:

$$\frac{\partial L}{\partial \mathbf{W}_C} = \frac{\partial L}{\partial C'}\frac{\partial C'}{\partial \mathbf{W}_C} = (\mathbf{W}_T\mathbf{h} + \mathbf{b}_T)\left(\frac{\partial L}{\partial C'}\right)^\top, \qquad \frac{\partial L}{\partial \mathbf{W}_T} = \frac{\partial L}{\partial C'}\frac{\partial C'}{\partial \mathbf{W}_T} = \mathbf{W}_C\frac{\partial L}{\partial C'}\mathbf{h}^\top. \tag{10}$$

For ease of understanding, the matrix of two equations are $\nabla_{\mathbf{W}_C}L = (\mathbf{W}_T\mathbf{h} + \mathbf{b}_T)\cdot(\nabla_{C'}L)^\top$ and $\nabla_{\mathbf{W}_T}L = \mathbf{W}_C\cdot\nabla_{C'}L\cdot\mathbf{h}^\top$. Then, the gradient of $\mathbf{b}_T$ is $\nabla_{\mathbf{b}_T}L = \mathbf{W}_C\cdot\nabla_{C'}L$. For the classifier $C'$, we calculate the gradient of $\mathbf{W}_{C'}$ and $\mathbf{b}_{C'}$ using $\nabla_{\mathbf{W}_{C'}}L = \mathbf{h}\cdot(\nabla_{C'}L)^\top$ and $\nabla_{\mathbf{b}_{C'}}L = \nabla_{C'}L$. According to $\mathbf{W}_{C'} = \mathbf{W}_T^\top\mathbf{W}_C$ and $\mathbf{b}_{C'} = \mathbf{W}_C^\top\mathbf{b}_T$, we analyze the gradient propagation using the chain rule:

$$\begin{aligned}
\Delta\mathbf{W}_{C'} &= \mathbf{W}_T^\top\Delta\mathbf{W}_C + (\Delta\mathbf{W}_T)^\top\mathbf{W}_C \\
&= \mathbf{W}_T^\top\left(\eta(\mathbf{W}_T\mathbf{h} + \mathbf{b}_T)\cdot(\nabla_{C'}L)^\top\right) + \eta\mathbf{h}\,(\nabla_{C'}L)\,\mathbf{W}_C^\top\mathbf{W}_C \\
&= \eta\mathbf{h}\cdot(\nabla_{C'}L)^\top, \quad \text{when } \mathbf{W}_C^\top\mathbf{W}_C = \mathbf{I}_k \text{ and } \mathbf{W}_T^\top\mathbf{W}_T = \mathbf{I}_d.
\end{aligned} \tag{11}$$

For $\mathbf{b}_{C'}$, we have the following:

$$\begin{aligned}
\Delta\mathbf{b}_{C'} &= \mathbf{W}_C^\top\Delta\mathbf{b}_T + (\Delta\mathbf{W}_C)^\top\mathbf{b}_T \\
&= \eta\mathbf{W}_C^\top\mathbf{W}_C\nabla_{C'}L + \eta\left(\mathbf{W}_T\mathbf{h} + \mathbf{b}_T\right)^\top\mathbf{b}_T\nabla_{C'}L,
\end{aligned} \tag{12}$$

when $\mathbf{W}_C$ has full column rank and $\mathbf{b}_T$ is orthogonal to $\mathbf{W}_T\mathbf{h} + \mathbf{b}_T$, we can obtain $\Delta\mathbf{b}_{C'} = \eta\nabla_{C'}L$, which is consistent with the gradient of single linear classifier $C'$.  □

# B  Other Experiments and Detail Settings

## B.1  Experimental Setup

**Implementation details.** In our experiments, we use 2-layer `GCN` backbones for `DGI` and `GRACE`, and 2-layer `GAT` backbone for `GraphMAE`. For downstream prompt tuning, all classifiers employ 2-layer `MLPs`. We fine-tune all GPL baselines across all pretrained models. We train for 2000 epochs with early stopping (patience=20). Following the `ProG` [2] benchmark settings, we conduct $k$-shot sampling evaluations under both in-domain and cross-domain settings with $k \in \{1, 3, 5\}$. To ensure performance reliability, we perform 20 repeated runs for each of 5 fixed random seeds = { 42, 12345, 23344, 38108, 39788 }, reporting averaged results over 100 total trials. All of the experiments are conducted on a server with Xeon(R) Platinum 8352V CPU, 90GB of memory, an RTX 4090 graphics card, and 24GB of video memory. The detailed `GitHub` links for the various pre-trained models, GPL baselines, sampling, split, and evaluation settings used in our experiments are provided in Table 12, which can be used for future reference and reproducibility.

## B.2  Real-world datasets

We introduce the details of the 10 commonly used real-world datasets, including homophily and heterophily graphs as follows, and the statistics of these datasets are shown in Table 4.

- *Cora* [34], *CiteSeer* [34] and *PubMed* [34] are citation datasets, nodes represent papers, edges represent citation relationships. Each dimension in the feature corresponds to a word. Labels are the categories into which the paper is divided.

- *Cornell* [35], *Texas* [35], and *Wisconsin* [35] are sub-datasets of WebKB [40], which is a webpage dataset collected from Carnegie Mellon University. Nodes represent web pages, and edges represent hyperlinks between web pages.

- *Chameleon* [35] and *Squirrel* [35] are page to page networks on specific topic collected from Wikipedia [41], nodes represent web pages and edges represent links between web pages. The average monthly traffic of the web page is converted into five categories to predict.

- *Actor* [35] is the actor-only induced subgraph of the film-director-actor-writer network. Each node corresponds to an actor, and the edge between two nodes denotes co-occurrence on the same Wikipedia page. Node features correspond to some keywords in the Wikipedia pages. The task is to classify the nodes into five categories in term of words of actor's Wikipedia.

- *arXiv-year* [42] is a modification of the OGBN-arXiv [43], where the labels are assigned based on the paper's publication year rather than topic. The nodes represent papers from arXiv website, and the links denote citation relationships. The node features are averaged Word2Vec [44] token features of both the title and abstract of the paper. The dataset is partitioned by publication date, which ensures a relatively balanced distribution of classes.

Table 4: Statistics of real-world datasets.

| Dataset | #Nodes | #Edges | #Features | #Classes | #Homophily |
|---|---|---|---|---|---|
| Cora | 2,708 | 5,278 | 1,433 | 7 | 0.81 |
| CiteSeer | 3,327 | 4,552 | 3,703 | 6 | 0.74 |
| PubMed | 19,717 | 44,324 | 500 | 3 | 0.80 |
| Cornell | 183 | 298 | 1,703 | 5 | 0.31 |
| Texas | 183 | 325 | 1,703 | 5 | 0.11 |
| Wisconsin | 251 | 515 | 1,703 | 5 | 0.20 |
| Chameleon | 2,277 | 36,101 | 2,277 | 5 | 0.24 |
| Actor | 7,600 | 30,019 | 932 | 5 | 0.22 |
| Squirrel | 5,201 | 217,073 | 2,089 | 5 | 0.22 |
| arXiv-year | 169,343 | 1,166,243 | 128 | 5 | 0.22 |

## B.3 Descriptions of Various Baselines

**Graph Semi-Supervised Baselines.**

- GCN [3]: GCN introduces a spectral graph convolution framework based on localized first-order Chebyshev filters, utilizing mean-pooling for neighborhood aggregation. It recursively updates node representations by averaging the features of neighbors and uses learnable parameters to control the transformation process.

- GAT [4]: GAT proposes multi-head attention mechanisms to dynamically compute node-specific weights during message passing. It adopts a learnable attention coefficient to quantify the importance of neighbors, thereby achieving adaptive aggregation.

**Graph Pretraining Models.** We introduce the classic graph pretraining strategies as follows.

- DGI [28]: Deep Graph Infomax (DGI) learns node embeddings by maximizing the mutual information (MI) between local node representations and graph representation. It utilizes GCNs to generate node representations, and aggregates node representations into a graph representation. DGI treats the corrupted graph as a negative example and train by identifying the relationship between nodes and graphs, thereby maximizing MI between them.

- GRACE [29]: GRACE learns node embeddings by maximizing mutual information between node representations in two augmented views. It generates different views through edge removal and feature masking. It uses InfoNCE [45, 46] as loss function, which maximizes the similarity of two augmented nodes generated by the same node and minimizes the similarity of other nodes to train the model.

Table 5: In-domain node classification. Accuracy on 3-shot node classification tasks over three pretrained strategies and nine datasets. The best results in each pretrain strategy are highlighted in **bold**, and the runner-up with an underline.

| Pretrain | Methods | Cora | CiteSeer | PubMed | Cornell | Texas | Wisconsin | Chameleon | Actor | Squirrel |
|---|---|---|---|---|---|---|---|---|---|---|
| DGI | Fine-tuning | $65.09_{\pm5.73}$ | $60.32_{\pm4.05}$ | $\underline{64.81}_{\pm6.80}$ | $\underline{41.84}_{\pm6.52}$ | $38.75_{\pm9.34}$ | $\underline{41.34}_{\pm7.57}$ | $\mathbf{28.66}_{\pm3.99}$ | $\underline{22.61}_{\pm2.36}$ | $23.02_{\pm3.54}$ |
| | Linear-probe | $67.48_{\pm4.65}$ | $\underline{60.91}_{\pm4.23}$ | $\mathbf{65.92}_{\pm5.53}$ | $40.39_{\pm8.35}$ | $39.30_{\pm7.67}$ | $38.29_{\pm9.18}$ | $\underline{27.17}_{\pm4.04}$ | $21.66_{\pm2.31}$ | $22.56_{\pm2.29}$ |
| | GPPT | $52.75_{\pm6.52}$ | $45.07_{\pm5.43}$ | $59.83_{\pm4.92}$ | $37.55_{\pm2.40}$ | $34.02_{\pm9.71}$ | $35.86_{\pm6.43}$ | $22.71_{\pm2.40}$ | $19.70_{\pm1.23}$ | $21.51_{\pm1.37}$ |
| | GraphPrompt | $64.29_{\pm47.8}$ | $57.37_{\pm5.26}$ | $60.56_{\pm5.37}$ | $26.06_{\pm6.24}$ | $36.89_{\pm7.56}$ | $25.96_{\pm9.75}$ | $25.71_{\pm2.68}$ | $20.02_{\pm1.39}$ | $22.16_{\pm2.42}$ |
| | All-in-one | $44.97_{\pm7.44}$ | $28.52_{\pm6.57}$ | $37.58_{\pm8.35}$ | $32.78_{\pm15.99}$ | $30.37_{\pm13.08}$ | $24.44_{\pm6.62}$ | $24.76_{\pm3.27}$ | $21.10_{\pm4.81}$ | $\underline{23.96}_{\pm2.92}$ |
| | GPF | $65.73_{\pm5.53}$ | $55.82_{\pm5.79}$ | $62.94_{\pm7.71}$ | $37.70_{\pm7.22}$ | $39.66_{\pm8.29}$ | $37.34_{\pm5.70}$ | $26.16_{\pm3.15}$ | $21.84_{\pm2.08}$ | $22.01_{\pm2.30}$ |
| | GPF+ | $\underline{68.17}_{\pm4.28}$ | $54.52_{\pm5.91}$ | $64.58_{\pm7.07}$ | $37.90_{\pm7.53}$ | $34.49_{\pm8.99}$ | $38.12_{\pm6.79}$ | $25.88_{\pm2.65}$ | $21.81_{\pm2.09}$ | $21.83_{\pm2.17}$ |
| | EdgePrompt | $62.65_{\pm3.39}$ | $49.49_{\pm4.97}$ | $59.56_{\pm3.43}$ | $40.26_{\pm7.81}$ | $41.88_{\pm8.76}$ | $36.59_{\pm7.37}$ | $25.07_{\pm4.07}$ | $21.63_{\pm2.45}$ | $21.45_{\pm1.83}$ |
| | EdgePrompt+ | $59.30_{\pm4.10}$ | $49.25_{\pm5.12}$ | $59.60_{\pm3.36}$ | $39.98_{\pm6.70}$ | $\underline{42.55}_{\pm9.04}$ | $37.01_{\pm7.58}$ | $25.45_{\pm3.77}$ | $22.39_{\pm2.80}$ | $21.96_{\pm1.87}$ |
| | UniPrompt(Ours) | $\mathbf{69.07}_{\pm4.37}$ | $\mathbf{61.73}_{\pm4.15}$ | $60.94_{\pm6.46}$ | $\mathbf{59.63}_{\pm5.84}$ | $\mathbf{61.44}_{\pm14.55}$ | $\mathbf{68.70}_{\pm6.99}$ | $25.90_{\pm3.08}$ | $\mathbf{27.32}_{\pm3.26}$ | $\mathbf{24.19}_{\pm2.35}$ |
| GRACE | Fine-tuning | $63.99_{\pm5.69}$ | $60.48_{\pm4.52}$ | $62.03_{\pm6.10}$ | $42.42_{\pm7.46}$ | $39.73_{\pm6.45}$ | $40.34_{\pm5.37}$ | $29.73_{\pm4.02}$ | $\underline{21.70}_{\pm2.14}$ | $23.77_{\pm2.23}$ |
| | Linear-probe | $63.68_{\pm5.99}$ | $60.35_{\pm3.99}$ | $\underline{65.71}_{\pm5.87}$ | $41.60_{\pm7.19}$ | $39.53_{\pm6.62}$ | $\underline{41.46}_{\pm5.28}$ | $29.02_{\pm4.60}$ | $21.55_{\pm1.38}$ | $21.62_{\pm3.47}$ |
| | GPPT | $54.24_{\pm7.29}$ | $51.71_{\pm5.39}$ | $57.13_{\pm4.60}$ | $36.05_{\pm8.71}$ | $33.55_{\pm3.66}$ | $37.69_{\pm5.65}$ | $31.45_{\pm3.69}$ | $20.78_{\pm1.21}$ | $24.17_{\pm2.68}$ |
| | GraphPrompt | $\underline{67.60}_{\pm4.90}$ | $53.84_{\pm8.22}$ | $56.60_{\pm7.00}$ | $29.86_{\pm7.56}$ | $34.82_{\pm8.84}$ | $29.66_{\pm8.12}$ | $25.40_{\pm3.18}$ | $20.98_{\pm1.85}$ | $\underline{24.41}_{\pm3.18}$ |
| | All-in-one | $40.78_{\pm8.81}$ | $31.09_{\pm5.80}$ | $38.25_{\pm6.23}$ | $30.60_{\pm17.45}$ | $30.66_{\pm12.45}$ | $24.40_{\pm16.68}$ | $25.40_{\pm3.18}$ | $21.32_{\pm2.56}$ | $23.50_{\pm2.32}$ |
| | GPF | $64.09_{\pm4.04}$ | $52.45_{\pm4.90}$ | $61.93_{\pm6.95}$ | $38.75_{\pm7.05}$ | $41.76_{\pm9.79}$ | $36.11_{\pm3.65}$ | $30.23_{\pm3.41}$ | $21.61_{\pm2.07}$ | $21.28_{\pm3.88}$ |
| | GPF+ | $63.91_{\pm5.08}$ | $53.24_{\pm6.88}$ | $55.88_{\pm5.54}$ | $38.40_{\pm5.00}$ | $41.37_{\pm8.39}$ | $36.63_{\pm5.07}$ | $\underline{32.07}_{\pm3.59}$ | $19.67_{\pm3.47}$ | $23.32_{\pm2.66}$ |
| | EdgePrompt | $60.45_{\pm4.39}$ | $48.65_{\pm4.08}$ | $57.33_{\pm4.67}$ | $\underline{42.58}_{\pm10.88}$ | $42.97_{\pm6.22}$ | $36.46_{\pm8.73}$ | $27.42_{\pm3.19}$ | $21.63_{\pm1.33}$ | $23.14_{\pm2.02}$ |
| | EdgePrompt+ | $61.60_{\pm2.95}$ | $45.12_{\pm4.53}$ | $62.38_{\pm6.11}$ | $42.11_{\pm9.13}$ | $\underline{43.83}_{\pm7.29}$ | $39.26_{\pm5.16}$ | $27.82_{\pm2.07}$ | $21.41_{\pm0.98}$ | $23.37_{\pm1.53}$ |
| | UniPrompt(Ours) | $\mathbf{67.71}_{\pm5.24}$ | $\mathbf{61.93}_{\pm3.73}$ | $\mathbf{66.83}_{\pm6.14}$ | $\mathbf{60.86}_{\pm8.37}$ | $\mathbf{64.22}_{\pm3.84}$ | $\mathbf{67.60}_{\pm8.57}$ | $27.71_{\pm3.66}$ | $\mathbf{25.56}_{\pm1.37}$ | $\mathbf{25.22}_{\pm2.47}$ |
| GraphMAE | Fine-tuning | $66.38_{\pm6.34}$ | $58.57_{\pm5.82}$ | $62.51_{\pm4.55}$ | $\underline{46.09}_{\pm8.50}$ | $\underline{43.91}_{\pm8.88}$ | $\underline{48.31}_{\pm5.70}$ | $27.33_{\pm3.17}$ | $21.40_{\pm1.56}$ | $21.18_{\pm1.20}$ |
| | Linear-probe | $\mathbf{70.74}_{\pm4.52}$ | $\underline{60.60}_{\pm4.96}$ | $\mathbf{66.90}_{\pm4.70}$ | $38.52_{\pm7.65}$ | $43.13_{\pm8.47}$ | $41.40_{\pm5.54}$ | $\mathbf{29.02}_{\pm4.05}$ | $\underline{22.08}_{\pm1.77}$ | $21.91_{\pm1.74}$ |
| | GPPT | $57.64_{\pm5.74}$ | $40.14_{\pm6.89}$ | $56.63_{\pm8.23}$ | $35.12_{\pm9.70}$ | $38.28_{\pm9.54}$ | $40.94_{\pm6.23}$ | $27.46_{\pm2.27}$ | $20.06_{\pm2.56}$ | $20.58_{\pm1.13}$ |
| | GraphPrompt | $\underline{67.49}_{\pm3.01}$ | $57.51_{\pm6.52}$ | $62.78_{\pm4.76}$ | $23.79_{\pm7.02}$ | $29.76_{\pm10.51}$ | $27.90_{\pm8.98}$ | $23.02_{\pm3.37}$ | $21.50_{\pm2.05}$ | $\mathbf{26.29}_{\pm2.34}$ |
| | All-in-one | $39.30_{\pm6.31}$ | $39.39_{\pm3.38}$ | $54.72_{\pm10.13}$ | $29.82_{\pm7.23}$ | $24.80_{\pm14.33}$ | $26.93_{\pm14.46}$ | $24.40_{\pm3.76}$ | $21.13_{\pm2.23}$ | $22.16_{\pm3.42}$ |
| | GPF | $57.91_{\pm4.28}$ | $43.44_{\pm12.02}$ | $64.32_{\pm7.22}$ | $36.33_{\pm6.82}$ | $38.79_{\pm9.89}$ | $36.86_{\pm5.95}$ | $27.09_{\pm2.88}$ | $21.30_{\pm2.51}$ | $20.82_{\pm1.81}$ |
| | GPF+ | $56.55_{\pm6.74}$ | $44.71_{\pm6.36}$ | $60.60_{\pm7.87}$ | $38.59_{\pm7.84}$ | $37.27_{\pm8.10}$ | $38.06_{\pm9.06}$ | $26.87_{\pm3.29}$ | $20.56_{\pm3.21}$ | $20.95_{\pm0.95}$ |
| | EdgePrompt | $64.18_{\pm4.20}$ | $57.56_{\pm6.66}$ | $54.32_{\pm7.07}$ | $35.42_{\pm6.17}$ | $40.95_{\pm8.73}$ | $37.31_{\pm5.69}$ | $26.60_{\pm4.02}$ | $19.66_{\pm4.94}$ | $22.05_{\pm1.27}$ |
| | EdgePrompt+ | $64.36_{\pm3.89}$ | $53.46_{\pm6.12}$ | $63.05_{\pm6.35}$ | $37.20_{\pm6.09}$ | $41.00_{\pm8.92}$ | $38.80_{\pm5.81}$ | $22.10_{\pm2.67}$ | $20.59_{\pm4.43}$ | $21.72_{\pm0.90}$ |
| | UniPrompt(Ours) | $66.16_{\pm6.69}$ | $\mathbf{61.90}_{\pm2.95x}$ | $\underline{64.62}_{\pm5.71}$ | $\mathbf{59.92}_{\pm5.06}$ | $\mathbf{65.62}_{\pm2.75}$ | $\mathbf{71.60}_{\pm2.88}$ | $\underline{27.78}_{\pm2.17}$ | $\mathbf{24.77}_{\pm1.83}$ | $\underline{22.82}_{\pm1.19}$ |

- GraphCL [37]: `GraphCL` learns graph-level representations by maximizing mutual information between augmented views of graphs. It introduces four graph augmentation types (node dropping, edge perturbation, attribute masking, subgraph sampling) to generate augmented views. The InfoNCE loss maximizes similarities between positive pairs (augmented views of the same graph) while contrasting against negative pairs (other graphs in the batch), corresponding to mutual information maximization between augmented representations and unifies diverse contrastive learning frameworks.

- GraphMAE [30]: `GraphMAE` is a generative self-supervised graph autoencoder that learns robust representations through masked feature reconstruction. It employs a two-stage framework: (1) A GNN-based encoder learns node embeddings from input graphs with randomly masked node features; (2) A GNN decoder reconstructs the masked features using a re-mask decoding strategy, optimized by a scaled cosine error loss that emphasizes directional alignment over magnitude.

**Graph Prompt Learning Baselines.**

- GPPT [6]: `GPPT` pioneers the use of the "pretrain–prompt" paradigm in graph machine learning. It employs link prediction as its pre-training task to learn general knowledge about graph structures. In its prompt design, `GPPT` introduces two types of prompts: task tokens and structure tokens. The former serve as prototype vectors for each class within clusters, while the latter are derived by aggregating information from the target node and its neighbors. For downstream node classification, the task is reformulated as link prediction. This is accomplished by calculating the probability of a link existing between the task token and the structure token, thus leveraging the learned prompts to make predictions.

- GraphPrompt [24]: `GraphPrompt` proposes a unified pretraining and prompting framework for GNNs, bridging the gap between pretraining and downstream tasks through a subgraph similarity-based template. It introduces learnable task-specific prompts that guide the ReadOut operation to dynamically emphasize task-relevant features during subgraph representation aggregation. By mapping both link prediction (pretraining) and node/graph classification (downstream) tasks to subgraph similarity learning, `GraphPrompt` enables parameter-efficient adaptation via prompt tuning—freezing pre-trained GNN weights while optimizing lightweight prompts.

- All-in-one [14] `All-in-one` unifies the downstream tasks of the "pretrain-prompt" paradigm. This method first reformulates node and edge tasks into graph-level tasks by

Table 6: In-domain node classification. Accuracy on 5-shot node classification tasks over three pretrained models and nine datasets. The best results in each pretrain strategy are highlighted in **bold**, and the runner-up with an underline.

| Pretrain | Methods | Cora | CiteSeer | PubMed | Cornell | Texas | Wisconsin | Chameleon | Actor | Squirrel |
|---|---|---|---|---|---|---|---|---|---|---|
| DGI | Fine-tuning | $\mathbf{73.01}_{\pm2.55}$ | $65.08_{\pm3.52}$ | $\underline{70.91}_{\pm4.65}$ | $\underline{45.78}_{\pm5.65}$ | $43.20_{\pm9.51}$ | $43.26_{\pm7.43}$ | $\mathbf{28.81}_{\pm2.82}$ | $23.65_{\pm2.38}$ | $22.58_{\pm2.75}$ |
| | Linear-probe | $\underline{72.39}_{\pm2.01}$ | $\mathbf{65.11}_{\pm2.62}$ | $70.32_{\pm4.19}$ | $45.23_{\pm6.87}$ | $42.81_{\pm8.09}$ | $41.66_{\pm5.68}$ | $\underline{28.80}_{\pm2.67}$ | $22.55_{\pm2.40}$ | $23.53_{\pm1.70}$ |
| | GPPT | $57.78_{\pm4.46}$ | $51.64_{\pm5.06}$ | $64.59_{\pm3.68}$ | $41.95_{\pm4.57}$ | $42.19_{\pm6.56}$ | $41.37_{\pm5.85}$ | $23.47_{\pm2.98}$ | $20.87_{\pm1.24}$ | $21.80_{\pm1.47}$ |
| | GraphPrompt | $65.36_{\pm4.72}$ | $62.33_{\pm2.60}$ | $66.83_{\pm6.05}$ | $27.94_{\pm6.51}$ | $40.91_{\pm7.12}$ | $31.20_{\pm7.22}$ | $25.98_{\pm3.38}$ | $20.38_{\pm1.04}$ | $22.82_{\pm2.18}$ |
| | All-in-one | $45.79_{\pm8.06}$ | $28.43_{\pm3.39}$ | $41.32_{\pm6.26}$ | $34.02_{\pm8.02}$ | $32.29_{\pm15.12}$ | $30.76_{\pm18.03}$ | $23.50_{\pm3.52}$ | $20.60_{\pm3.11}$ | $\underline{23.78}_{\pm2.93}$ |
| | GPF | $66.57_{\pm7.50}$ | $60.99_{\pm3.73}$ | $68.33_{\pm5.03}$ | $42.96_{\pm6.01}$ | $42.61_{\pm8.83}$ | $43.68_{\pm6.29}$ | $27.10_{\pm2.94}$ | $22.79_{\pm1.56}$ | $23.38_{\pm2.37}$ |
| | GPF+ | $69.10_{\pm3.70}$ | $57.84_{\pm4.22}$ | $68.81_{\pm4.57}$ | $43.63_{\pm6.62}$ | $43.21_{\pm8.64}$ | $45.11_{\pm6.42}$ | $27.86_{\pm2.74}$ | $22.39_{\pm2.01}$ | $21.48_{\pm3.01}$ |
| | EdgePrompt | $66.82_{\pm3.62}$ | $56.99_{\pm4.12}$ | $64.08_{\pm6.27}$ | $45.14_{\pm5.71}$ | $\underline{49.10}_{\pm11.77}$ | $47.61_{\pm6.32}$ | $25.05_{\pm3.76}$ | $\underline{23.82}_{\pm1.87}$ | $21.62_{\pm1.53}$ |
| | EdgePrompt+ | $67.10_{\pm3.94}$ | $56.12_{\pm3.88}$ | $62.95_{\pm6.15}$ | $43.05_{\pm4.58}$ | $46.88_{\pm9.06}$ | $\underline{50.40}_{\pm5.49}$ | $24.96_{\pm3.74}$ | $23.49_{\pm1.99}$ | $21.53_{\pm2.15}$ |
| | UniPrompt(Ours) | $70.58_{\pm3.01}$ | $\underline{65.10}_{\pm3.15}$ | $\mathbf{70.97}_{\pm4.33}$ | $\mathbf{68.02}_{\pm4.32}$ | $\mathbf{67.86}_{\pm8.36}$ | $\mathbf{70.43}_{\pm9.34}$ | $28.04_{\pm2.68}$ | $\mathbf{28.20}_{\pm2.66}$ | $\mathbf{23.88}_{\pm2.19}$ |
| GRACE | Fine-tuning | $70.49_{\pm2.28}$ | $\mathbf{64.19}_{\pm3.49}$ | $70.42_{\pm5.36}$ | $47.15_{\pm6.77}$ | $43.09_{\pm8.74}$ | $42.51_{\pm5.92}$ | $\mathbf{34.00}_{\pm2.48}$ | $22.61_{\pm1.91}$ | $\underline{25.22}_{\pm1.65}$ |
| | Linear-probe | $\underline{71.09}_{\pm2.18}$ | $\underline{63.65}_{\pm3.29}$ | $\underline{71.34}_{\pm6.46}$ | $47.07_{\pm6.66}$ | $42.11_{\pm8.02}$ | $41.91_{\pm6.20}$ | $32.78_{\pm3.15}$ | $22.23_{\pm1.88}$ | $24.05_{\pm1.58}$ |
| | GPPT | $56.51_{\pm7.10}$ | $50.88_{\pm5.62}$ | $65.97_{\pm5.75}$ | $44.36_{\pm4.88}$ | $41.15_{\pm7.09}$ | $41.98_{\pm7.60}$ | $33.10_{\pm3.47}$ | $21.36_{\pm2.18}$ | $24.70_{\pm2.14}$ |
| | GraphPrompt | $68.58_{\pm4.30}$ | $52.65_{\pm3.84}$ | $65.49_{\pm6.66}$ | $35.28_{\pm5.94}$ | $38.65_{\pm7.75}$ | $29.49_{\pm16.00}$ | $23.77_{\pm3.65}$ | $21.17_{\pm1.14}$ | $22.55_{\pm1.87}$ |
| | All-in-one | $44.29_{\pm8.37}$ | $39.27_{\pm3.85}$ | $40.76_{\pm8.25}$ | $29.23_{\pm7.65}$ | $30.71_{\pm10.37}$ | $29.49_{\pm16.00}$ | $23.77_{\pm3.65}$ | $21.57_{\pm2.49}$ | $25.13_{\pm2.82}$ |
| | GPF | $68.56_{\pm3.98}$ | $59.53_{\pm3.91}$ | $68.20_{\pm4.59}$ | $46.01_{\pm6.72}$ | $44.17_{\pm7.43}$ | $41.66_{\pm4.84}$ | $28.62_{\pm3.26}$ | $22.91_{\pm1.49}$ | $21.29_{\pm2.57}$ |
| | GPF+ | $68.86_{\pm3.95}$ | $61.51_{\pm3.90}$ | $68.30_{\pm3.99}$ | $45.89_{\pm6.10}$ | $40.99_{\pm8.35}$ | $45.65_{\pm5.22}$ | $29.46_{\pm3.33}$ | $22.64_{\pm1.45}$ | $24.61_{\pm2.24}$ |
| | EdgePrompt | $63.76_{\pm3.49}$ | $51.81_{\pm6.08}$ | $68.23_{\pm3.16}$ | $\underline{48.17}_{\pm8.16}$ | $\mathbf{54.45}_{\pm7.50}$ | $\underline{47.14}_{\pm6.15}$ | $32.04_{\pm3.80}$ | $23.17_{\pm1.55}$ | $24.22_{\pm1.26}$ |
| | EdgePrompt+ | $66.16_{\pm3.38}$ | $53.90_{\pm3.03}$ | $71.03_{\pm2.40}$ | $47.66_{\pm8.64}$ | $53.98_{\pm6.93}$ | $46.46_{\pm5.98}$ | $32.44_{\pm3.41}$ | $\underline{24.46}_{\pm1.64}$ | $24.35_{\pm1.41}$ |
| | UniPrompt(Ours) | $72.99_{\pm3.48}$ | $63.64_{\pm3.80}$ | $\mathbf{74.21}_{\pm2.81}$ | $68.13_{\pm4.35}$ | $\mathbf{68.36}_{\pm4.92}$ | $\mathbf{71.43}_{\pm4.58}$ | $\underline{33.66}_{\pm1.54}$ | $\mathbf{26.68}_{\pm1.87}$ | $\mathbf{26.07}_{\pm0.84}$ |
| GraphMAE | Fine-tuning | $73.85_{\pm2.87}$ | $64.59_{\pm4.32}$ | $\mathbf{72.83}_{\pm3.21}$ | $\underline{58.24}_{\pm4.44}$ | $\underline{47.62}_{\pm6.96}$ | $50.29_{\pm6.59}$ | $28.78_{\pm2.47}$ | $21.22_{\pm4.17}$ | $22.38_{\pm1.12}$ |
| | Linear-probe | $\mathbf{75.78}_{\pm2.38}$ | $\mathbf{66.17}_{\pm2.72}$ | $70.08_{\pm4.82}$ | $43.71_{\pm5.71}$ | $45.00_{\pm7.98}$ | $41.11_{\pm7.54}$ | $\mathbf{31.31}_{\pm3.63}$ | $22.51_{\pm2.23}$ | $22.25_{\pm1.75}$ |
| | GPPT | $64.66_{\pm5.47}$ | $46.87_{\pm6.52}$ | $62.50_{\pm7.81}$ | $46.25_{\pm4.64}$ | $41.04_{\pm6.81}$ | $46.10_{\pm5.06}$ | $26.49_{\pm3.32}$ | $20.14_{\pm2.49}$ | $20.97_{\pm1.15}$ |
| | GraphPrompt | $69.80_{\pm4.65}$ | $49.17_{\pm4.19}$ | $67.51_{\pm6.93}$ | $25.83_{\pm5.77}$ | $38.54_{\pm9.35}$ | $30.60_{\pm7.55}$ | $23.65_{\pm2.67}$ | $20.08_{\pm1.65}$ | $\mathbf{24.71}_{\pm2.50}$ |
| | All-in-one | $41.06_{\pm6.34}$ | $41.97_{\pm3.61}$ | $63.56_{\pm5.10}$ | $27.98_{\pm8.33}$ | $24.11_{\pm8.95}$ | $31.56_{\pm17.46}$ | $23.53_{\pm3.43}$ | $21.60_{\pm2.78}$ | $23.19_{\pm3.54}$ |
| | GPF | $72.09_{\pm3.98}$ | $52.73_{\pm4.94}$ | $65.47_{\pm4.85}$ | $41.55_{\pm6.06}$ | $43.08_{\pm8.13}$ | $42.30_{\pm5.95}$ | $28.51_{\pm2.63}$ | $\underline{22.62}_{\pm3.36}$ | $21.04_{\pm1.39}$ |
| | GPF+ | $63.28_{\pm6.20}$ | $55.60_{\pm6.03}$ | $65.96_{\pm5.03}$ | $44.53_{\pm7.08}$ | $39.61_{\pm6.69}$ | $42.38_{\pm6.35}$ | $26.86_{\pm3.34}$ | $20.89_{\pm2.70}$ | $21.12_{\pm0.81}$ |
| | EdgePrompt | $67.74_{\pm3.60}$ | $62.29_{\pm3.24}$ | $58.66_{\pm6.36}$ | $44.37_{\pm6.27}$ | $44.02_{\pm7.88}$ | $43.53_{\pm5.40}$ | $29.35_{\pm2.20}$ | $22.00_{\pm2.67}$ | $21.68_{\pm1.32}$ |
| | EdgePrompt+ | $73.81_{\pm2.00}$ | $48.25_{\pm4.39}$ | $65.60_{\pm3.68}$ | $44.02_{\pm7.48}$ | $44.56_{\pm7.51}$ | $43.14_{\pm5.61}$ | $\underline{29.84}_{\pm2.28}$ | $21.74_{\pm2.57}$ | $21.35_{\pm1.15}$ |
| | UniPrompt(Ours) | $\underline{74.77}_{\pm2.26}$ | $\underline{65.74}_{\pm2.80}$ | $\underline{70.49}_{\pm4.77}$ | $\mathbf{67.73}_{\pm3.71}$ | $\mathbf{71.02}_{\pm5.21}$ | $\mathbf{73.89}_{\pm6.62}$ | $29.77_{\pm2.26}$ | $\mathbf{24.96}_{\pm1.65}$ | $\underline{23.23}_{\pm1.21}$ |

constructing an induced subgraph. It then generates learnable prompts and integrates them into the node features in a weighted manner to construct a prompt graph. Furthermore, this method combines meta-learning to optimize prompts across multiple tasks and make the prompts adapt to different downstream tasks.

- GPF/GPF-plus [7]: GPF/GPF-plus framework introduces a universal graph prompt learning method that is compatible with any pre-training strategy. The approach operates by adding learnable prompt vectors to the input node features. Specifically, GPF uses a single global prompt vector shared by all nodes, whereas GPF-plus generates individual prompt vectors for each node by aggregating basis vectors via an attention mechanism. The resulting prompted nodes are then fed into a frozen pre-trained GNN for the downstream task. This method effectively overcomes a key limitation of existing prompt-tuning methods, which are restricted to specific pre-training tasks.

- EdgePrompt/EdgePrompt-plus [23]: EdgePrompt introduces a graph prompt tuning framework for pre-trained GNNs by injecting learnable edge-wise prompts into adjacency matrices. It designs edge-specific trainable vectors to customize message aggregation patterns between nodes. This structural adaptation bridges the objective gap between pretraining and downstream tasks while preserving GNN parameters. EdgePrompt+ enables each edge to learn its customized prompt vectors, which is similar to GPF and GPF-plus.

**Multi-Domain Graph Pretrain Baselines.**

- GCOPE [38]: GCOPE proposes a cross-domain graph pretraining framework that unifies diverse graph structures by introducing learnable "coordinators" to align various datasets. These coordinators interconnect isolated source datasets into a unified large-scale graph, enabling joint pretraining with objectives. During pretraining, GCOPE learns transferable representations by balancing shared multi-domain knowledge and domain-specific features through latent alignment strategies. The framework supports flexible transfer via fine-tuning or graph prompting while maintaining parameter efficiency.

- MDGPT [39]: MDGPT introduces a dual-prompt framework for downstream adaptation: a unifying prompt transfers broadly learned cross-domain knowledge by aligning target domains with the pre-trained prior, and a mixing prompt enables fine-grained domain-specific alignment through learnable projections. MDGPT bridges pretraining and downstream tasks by optimizing domain-invariant representations via self-supervised objectives on multi-domain data.

Table 7: Analysis of key components in `UniPrompt` via replacement experiments on 1-shot, 3-shot and 5-shot node classification tasks over different pretrained models.

| Shot | Pretrain | Strategies | Cora | CiteSeer | PubMed | Cornell | Texas | Wisconsin | Chameleon | Actor | Squirrel |
|---|---|---|---|---|---|---|---|---|---|---|---|
| 1 | DGI | Random_Topo | $45.81_{\pm8.86}$ | $40.81_{\pm10.22}$ | $62.84_{\pm4.16}$ | $32.34_{\pm14.79}$ | $22.03_{\pm15.34}$ | $34.74_{\pm7.21}$ | $21.82_{\pm2.17}$ | $21.46_{\pm3.05}$ | $23.83_{\pm1.52}$ |
| | | Simple_Add | $24.23_{\pm5.36}$ | $26.61_{\pm4.24}$ | $43.90_{\pm9.68}$ | $51.88_{\pm16.76}$ | $39.37_{\pm13.39}$ | $63.66_{\pm3.16}$ | $25.23_{\pm4.66}$ | $23.50_{\pm1.98}$ | $24.08_{\pm1.60}$ |
| | | Discard_Topo | $27.27_{\pm6.48}$ | $28.69_{\pm5.36}$ | $36.81_{\pm7.89}$ | $51.88_{\pm17.03}$ | $45.94_{\pm14.83}$ | $62.74_{\pm10.70}$ | $23.98_{\pm4.91}$ | $26.93_{\pm3.48}$ | $23.37_{\pm1.09}$ |
| | GRACE | Random_Topo | $40.48_{\pm8.03}$ | $17.28_{\pm0.55}$ | $66.58_{\pm6.08}$ | $29.69_{\pm9.96}$ | $26.72_{\pm3.47}$ | $27.54_{\pm7.58}$ | $27.35_{\pm5.48}$ | $20.63_{\pm1.11}$ | $24.25_{\pm2.86}$ |
| | | Simple_Add | $39.04_{\pm10.74}$ | $15.70_{\pm1.84}$ | $61.60_{\pm5.58}$ | $60.16_{\pm6.50}$ | $21.56_{\pm12.82}$ | $49.83_{\pm11.60}$ | $23.32_{\pm1.41}$ | $23.29_{\pm2.25}$ | $23.79_{\pm1.86}$ |
| | | Discard_Topo | $39.20_{\pm11.25}$ | $16.13_{\pm2.58}$ | $39.25_{\pm2.03}$ | $58.44_{\pm9.06}$ | $29.84_{\pm19.27}$ | $51.89_{\pm12.15}$ | | $26.48_{\pm3.83}$ | $23.85_{\pm1.59}$ |
| | GraphMAE | Random_Topo | $38.58_{\pm6.42}$ | $33.77_{\pm9.37}$ | $41.02_{\pm5.34}$ | $23.13_{\pm7.24}$ | $34.53_{\pm12.78}$ | $31.20_{\pm4.10}$ | $20.15_{\pm1.41}$ | $19.30_{\pm3.23}$ | $20.94_{\pm0.22}$ |
| | | Simple_Add | $49.11_{\pm9.59}$ | $51.46_{\pm11.88}$ | $57.79_{\pm9.90}$ | $49.38_{\pm11.01}$ | $42.97_{\pm14.82}$ | $65.26_{\pm13.60}$ | $25.47_{\pm1.71}$ | $21.86_{\pm3.21}$ | $23.06_{\pm1.73}$ |
| | | Discard_Topo | $46.01_{\pm9.86}$ | $49.99_{\pm11.70}$ | $36.83_{\pm7.48}$ | $49.69_{\pm13.36}$ | $44.69_{\pm15.52}$ | $63.43_{\pm13.45}$ | $25.40_{\pm4.06}$ | $21.77_{\pm4.99}$ | $23.19_{\pm2.11}$ |
| 3 | DGI | Random_Topo | $66.20_{\pm2.61}$ | $60.51_{\pm3.54}$ | $65.50_{\pm2.87}$ | $38.28_{\pm2.47}$ | $25.62_{\pm3.93}$ | $34.74_{\pm8.00}$ | $23.71_{\pm2.09}$ | $19.36_{\pm2.10}$ | $19.93_{\pm0.53}$ |
| | | Simple_Add | $26.20_{\pm5.33}$ | $28.24_{\pm6.10}$ | $57.75_{\pm4.46}$ | $63.44_{\pm2.72}$ | $34.22_{\pm19.07}$ | $61.83_{\pm7.44}$ | $24.97_{\pm1.96}$ | $25.30_{\pm1.44}$ | $22.74_{\pm2.01}$ |
| | | Discard_Topo | $50.87_{\pm7.24}$ | $45.95_{\pm6.91}$ | $59.03_{\pm5.21}$ | $62.19_{\pm2.55}$ | $55.63_{\pm17.81}$ | $69.60_{\pm10.99}$ | $26.73_{\pm2.44}$ | $27.71_{\pm1.33}$ | $21.12_{\pm2.55}$ |
| | GRACE | Random_Topo | $64.78_{\pm8.27}$ | $60.13_{\pm3.83}$ | $69.28_{\pm5.93}$ | $40.47_{\pm5.44}$ | $34.06_{\pm5.75}$ | $30.29_{\pm5.27}$ | $22.65_{\pm2.55}$ | $21.35_{\pm1.16}$ | $25.02_{\pm0.68}$ |
| | | Simple_Add | $53.64_{\pm4.69}$ | $48.61_{\pm5.15}$ | $61.37_{\pm5.95}$ | $54.06_{\pm11.43}$ | $31.41_{\pm19.65}$ | $65.60_{\pm5.38}$ | $26.76_{\pm3.53}$ | $23.97_{\pm1.39}$ | $24.83_{\pm1.21}$ |
| | | Discard_Topo | $53.71_{\pm5.12}$ | $49.35_{\pm4.45}$ | $36.27_{\pm9.04}$ | $54.22_{\pm12.55}$ | $65.94_{\pm6.57}$ | $66.29_{\pm7.11}$ | $28.01_{\pm2.28}$ | $24.78_{\pm1.49}$ | $25.20_{\pm1.51}$ |
| | GraphMAE | Random_Topo | $55.34_{\pm2.86}$ | $41.25_{\pm5.71}$ | $51.16_{\pm2.39}$ | $33.75_{\pm7.03}$ | $30.00_{\pm7.26}$ | $28.11_{\pm8.42}$ | $21.60_{\pm1.62}$ | $21.31_{\pm1.72}$ | $20.42_{\pm0.69}$ |
| | | Simple_Add | $69.77_{\pm5.32}$ | $63.56_{\pm1.45}$ | $66.22_{\pm5.05}$ | $59.84_{\pm5.58}$ | $37.50_{\pm11.69}$ | $58.29_{\pm12.59}$ | $24.81_{\pm1.09}$ | $25.37_{\pm2.73}$ | $20.82_{\pm1.60}$ |
| | | Discard_Topo | $51.72_{\pm4.48}$ | $62.99_{\pm1.04}$ | $56.93_{\pm4.10}$ | $60.93_{\pm5.74}$ | $45.78_{\pm13.95}$ | $57.71_{\pm12.22}$ | $26.11_{\pm3.46}$ | $24.63_{\pm3.30}$ | $21.12_{\pm1.56}$ |
| 5 | DGI | Random_Topo | $55.75_{\pm3.37}$ | $63.27_{\pm1.41}$ | $72.11_{\pm1.45}$ | $44.37_{\pm5.69}$ | $36.09_{\pm4.72}$ | $40.80_{\pm6.40}$ | $23.37_{\pm1.75}$ | $21.73_{\pm2.11}$ | $19.34_{\pm1.05}$ |
| | | Simple_Add | $24.07_{\pm2.17}$ | $33.53_{\pm7.95}$ | $52.91_{\pm2.24}$ | $66.87_{\pm3.58}$ | $39.37_{\pm22.46}$ | $70.74_{\pm4.58}$ | $25.62_{\pm2.22}$ | $25.15_{\pm1.68}$ | $21.20_{\pm1.43}$ |
| | | Discard_Topo | $37.33_{\pm3.23}$ | $49.75_{\pm2.76}$ | $61.54_{\pm6.05}$ | $66.72_{\pm2.91}$ | $69.22_{\pm3.15}$ | $74.06_{\pm3.64}$ | $28.38_{\pm1.44}$ | $28.04_{\pm2.04}$ | $22.60_{\pm2.94}$ |
| | GRACE | Random_Topo | $69.53_{\pm1.76}$ | $64.02_{\pm2.20}$ | $69.79_{\pm5.09}$ | $46.56_{\pm4.98}$ | $37.66_{\pm4.12}$ | $41.37_{\pm6.74}$ | $33.66_{\pm1.80}$ | $23.19_{\pm0.66}$ | $25.11_{\pm1.88}$ |
| | | Simple_Add | $60.94_{\pm2.39}$ | $51.47_{\pm2.60}$ | $50.09_{\pm15.64}$ | $67.19_{\pm1.48}$ | $63.91_{\pm7.03}$ | $67.20_{\pm5.43}$ | $28.08_{\pm2.43}$ | $23.79_{\pm1.41}$ | $24.59_{\pm1.99}$ |
| | | Discard_Topo | $61.31_{\pm2.42}$ | $53.59_{\pm0.87}$ | $37.02_{\pm6.73}$ | $67.97_{\pm2.84}$ | $69.84_{\pm7.20}$ | $70.29_{\pm3.84}$ | $29.01_{\pm2.23}$ | $25.68_{\pm0.64}$ | $24.99_{\pm2.25}$ |
| | GraphMAE | Random_Topo | $47.71_{\pm6.19}$ | $46.10_{\pm2.35}$ | $57.95_{\pm4.81}$ | $38.59_{\pm2.73}$ | $39.37_{\pm5.10}$ | $38.63_{\pm3.66}$ | $22.02_{\pm1.63}$ | $19.99_{\pm0.64}$ | $20.90_{\pm0.54}$ |
| | | Simple_Add | $69.72_{\pm2.62}$ | $65.66_{\pm3.02}$ | $73.22_{\pm3.31}$ | $67.19_{\pm1.40}$ | $66.72_{\pm8.94}$ | $74.74_{\pm4.72}$ | $28.42_{\pm3.01}$ | $23.80_{\pm2.23}$ | $23.65_{\pm1.02}$ |
| | | Discard_Topo | $59.12_{\pm4.09}$ | $61.96_{\pm4.11}$ | $68.70_{\pm2.40}$ | $66.25_{\pm3.40}$ | $68.75_{\pm4.50}$ | $75.43_{\pm4.40}$ | $29.34_{\pm2.01}$ | $23.59_{\pm1.64}$ | $23.08_{\pm1.79}$ |

- MDGFM [27]: MDGFM integrates multiple source domains during pretraining, leveraging contrastive learning to maximize mutual information between multi-view graph augmentations. The topology-aware refinement process, which aligns different graph topologies into a unified semantic space via meta-prompts (for global knowledge transfer) and task-specific prompts (for domain adaptation).

### B.4  3/5-shot node classification on different pretrained models

We further conduct 3-shot and 5-shot node classification experiments on the same nine datasets, based on three different pretrained strategies, as shown in Table 5 and Table 6. Consistent with the 1-shot results, our method outperforms existing GPL approaches across most datasets under different pretrained model settings. However, the baselines become more competitive in these scenarios, with each achieving runner-up on certain datasets. Another observation is that, as the number of shots increases, the performance discrepancies among models under different pretrained settings become more pronounced, especially on datasets such as *CiteSeer* and *Chameleon*. In comparison to the GPL baselines, `UniPrompt` demonstrates more stable performance across all settings. It is also noteworthy that with an increase in the number of labels, `Fine-tuning` and `Linear-probe` become highly competitive, achieving runner-up or even optimal performance on many datasets. This indicates that traditional fine-tuning methods, much like GPLs, benefit significantly from additional label information and, in some instances, utilize it more effectively than some existing prompt approaches. Furthermore, our method achieves particularly significant gains on the *Cornell*, *Texas*, and *Wisconsin* datasets, with this strong performance holding consistently across the different pretrained models. This further underscores the broad applicability and robustness of `UniPrompt` .

### B.5  Key Components Analysis of UniPrompt

To further analyze the pros and cons of each component in `UniPrompt`, we initially aimed to remove the key components (i.e., $k$NN and bootstrap). However, since both components are essential and cannot be simply removed, we instead conduct replacement experiments: (1) `Random_Topo`: replacing $k$NN with random topology, (2) `Simple_Add`: replacing bootstrap with a simple addition of the original and prompt graph, and (3) `Discard_Topo`: discarding the original graph totally. The 1-shot results are shown in the Table 7: From the table, we can find that `Random_Topo` maintains some of effectiveness on homophilic datasets while showing reduced performance on heterophilic ones. Conversely, for `Simple_Add` and `Discard_Topo`, heterophilic datasets still retain some performance. However, performance on homophilic datasets drops significantly, as their original structure is crucial for classification. Furthermore, a notable phenomenon is that when these core

Table 8: In-domain large scale node classification. Accuracy on 5-shot node classification tasks over three pretrained models and *arXiv-year* dataset. The best result is highlighted in **bold**, and the runner-up with an underline.

| Pretrain | Method | arXiv-year (Acc) | Preprocessing Time (s) | Training Time (s/per_epoch) |
|---|---|---|---|---|
| DGI | Fine-tune | $\underline{28.27}_{\pm 5.99}$ | - | 0.0138 |
| | UniPrompt | $\mathbf{32.48}_{\pm 6.37}$ | 1.25 | 0.0224 |
| GRACE | Fine-tune | $24.60_{\pm 1.04}$ | - | 0.0205 |
| | UniPrompt | $25.17_{\pm 2.83}$ | 1.26 | 0.0320 |
| GraphMAE | Fine-tune | $23.24_{\pm 1.58}$ | - | 0.0427 |
| | UniPrompt | $24.25_{\pm 5.43}$ | 1.32 | 0.0618 |

Table 9: Time (s) and GPU memory (MB) costs of different GPL baselines across various datasets.

| Methods | Time/Memory | Cora | CiteSeer | PubMed | Cornell | Texas | Wisconsin | Chameleon | Actor | Squirrel |
|---|---|---|---|---|---|---|---|---|---|---|
| Fine-tune | Time | 0.0027 | 0.0026 | 0.0033 | 0.0026 | 0.0025 | 0.0031 | 0.0026 | 0.0027 | 0.0034 |
| | Memory | 80.5 | 153.1 | 310.3 | 26.6 | 26.6 | 28.4 | 118.4 | 155.4 | 368.9 |
| Linear-probe | Time | 0.0010 | 0.0009 | 0.0009 | 0.0009 | 0.0009 | 0.0009 | 0.0011 | 0.0009 | 0.0009 |
| | Memory | 56.9 | 127.5 | 120.1 | 26.4 | 26.4 | 27.6 | 70.5 | 86.9 | 122.0 |
| GPPT | Time | 0.0118 | 0.0117 | 0.0130 | 0.0123 | 0.0120 | 0.0119 | 0.0109 | 0.0051 | 0.0127 |
| | Memory | 80.6 | 153.1 | 311.7 | 26.6 | 26.7 | 28.5 | 118.4 | 155.1 | 365.9 |
| GraphPrompt | Time | 0.0035 | 0.0008 | 0.0101 | 0.0242 | 0.0037 | 0.0136 | 0.0003 | 0.0006 | 0.0024 |
| | Memory | 642.6 | 896.1 | 2366.2 | 31.5 | 31.8 | 37.0 | 1170.4 | 913.7 | 3597.7 |
| All-in-one | Time | 0.7616 | 0.7362 | 0.6024 | 0.6846 | 0.6732 | 0.6846 | 0.6856 | 0.7118 | 0.6834 |
| | Memory | 2696.0 | 4021.5 | 9052.5 | 31.0 | 32.2 | 39.8 | 1801.8 | 3625.9 | 4917.2 |
| GPF | Time | 0.0021 | 0.0031 | 0.0031 | 0.0020 | 0.0021 | 0.0020 | 0.0031 | 0.0022 | 0.0042 |
| | Memory | 119.8 | 267.4 | 470.1 | 29.9 | 29.9 | 32.1 | 193.4 | 236.8 | 662.4 |
| GPF+ | Time | 0.0033 | 0.0032 | 0.0033 | 0.0021 | 0.0023 | 0.0021 | 0.0032 | 0.0022 | 0.0042 |
| | Memory | 130.0 | 361.8 | 470.5 | 32.3 | 32.3 | 34.9 | 194.3 | 236.9 | 662.6 |
| EdgePrompt | Time | 0.0018 | 0.0027 | 0.0042 | 0.0025 | 0.0017 | 0.0017 | 0.0041 | 0.0031 | 0.0127 |
| | Memory | 191.1 | 398.0 | 746.2 | 32.6 | 31.9 | 36.3 | 550.8 | 379.8 | 2608.3 |
| EdgePrompt+ | Time | 0.0024 | 0.0027 | 0.0042 | 0.0025 | 0.0024 | 0.0024 | 0.0040 | 0.0035 | 0.0122 |
| | Memory | 191.5 | 398.3 | 746.2 | 33.2 | 32.0 | 36.7 | 551.0 | 380.8 | 2608.4 |
| UniPrompt(Ours) | Time | 0.0054 | 0.0052 | 0.0039 | 0.0040 | 0.0045 | 0.0039 | 0.0047 | 0.0068 | 0.0073 |
| | Memory | 511.5 | 674.5 | 566.2 | 55.1 | 55.2 | 67.7 | 539.9 | 1392.6 | 1603.9 |

components are replaced, an increasing number of labels does not consistently lead to performance improvements. This is particularly evident in the `Random_Topo` setting. Although this setup is analogous to augmentation strategies [47, 37] in graph self-supervised learning, adding edges, as opposed to masking them, can introduce unnecessary message passing and additional potential risks. Thus, simply using a random topology is clearly suboptimal. Another observation is the varied impact of discarding the original topology across different datasets. On homophilic graphs, performance drops significantly, whereas in heterophilic scenarios, performance is maintained on some datasets (i.e., *Cornell*, *Texas*) or even improved (i.e., *Wisconsin*, *Chameleon*). This suggests that the original topology in these latter cases fails to provide an effective message passing mechanism, and useful information is instead derived by learning the distribution of representations around anchor nodes. In contrast, `UniPrompt` augments the graph with a learnable topology, facilitating effective message passing for downstream adaptation and thereby ensuring good performance on both homophilic and heterophilic datasets.

## B.6  Large Scale Dataset Node Classification

We additionally run experiments on the large-scale heterophilic dataset *Arxiv-year* as a supplement. Here, we use a simplified $k$NN by randomly sampling 1,000 nodes, then connecting each node to its top-$k$ most similar sampled nodes. We test three pretrain strategies under 5-shot setting, comparing with fine-tuning. The accuracy and computational cost are shown in the Table 8. Our method incurs minimal preprocessing time and only a slight increase in training time per epoch, with small epoch counts (typically less than 500). This demonstrates that our approach is scalable to large graphs.

Table 10: Robustness analysis of the various pre-trained models to varying levels of Gaussian noise on 1-shot, 3-shot, 5-shot node classification.

| Pretrain | Shot | Noisy | Cora | CiteSeer | PubMed | Cornell | Texas | Wisconsin | Chameleon | Actor | Squirrel |
|---|---|---|---|---|---|---|---|---|---|---|---|
| DGI | 1 | 0.01 | $44.42_{\pm10.49}$ | $32.39_{\pm12.85}$ | $61.00_{\pm5.31}$ | $50.62_{\pm14.07}$ | $46.72_{\pm12.88}$ | $62.51_{\pm10.50}$ | $20.80_{\pm2.43}$ | $26.96_{\pm4.35}$ | $22.50_{\pm2.07}$ |
| | | 0.05 | $23.23_{\pm9.21}$ | $19.77_{\pm1.66}$ | $40.01_{\pm0.98}$ | $49.22_{\pm12.02}$ | $38.75_{\pm13.32}$ | $61.03_{\pm9.79}$ | $20.72_{\pm0.76}$ | $24.67_{\pm2.96}$ | $20.41_{\pm0.74}$ |
| | | 0.20 | $27.82_{\pm5.26}$ | $15.73_{\pm5.90}$ | $39.46_{\pm0.37}$ | $28.59_{\pm5.96}$ | $33.91_{\pm10.64}$ | $42.86_{\pm16.34}$ | $20.08_{\pm2.77}$ | $22.15_{\pm2.70}$ | $20.21_{\pm0.27}$ |
| | 3 | 0.01 | $63.72_{\pm4.62}$ | $27.40_{\pm6.12}$ | $63.33_{\pm4.47}$ | $52.66_{\pm1.82}$ | $45.31_{\pm19.04}$ | $56.80_{\pm8.75}$ | $23.27_{\pm2.51}$ | $24.58_{\pm2.75}$ | $21.50_{\pm1.19}$ |
| | | 0.05 | $18.65_{\pm9.51}$ | $16.23_{\pm4.72}$ | $39.72_{\pm4.32}$ | $56.87_{\pm4.62}$ | $45.16_{\pm20.96}$ | $55.54_{\pm6.71}$ | $19.39_{\pm0.59}$ | $24.47_{\pm3.06}$ | $20.08_{\pm0.21}$ |
| | | 0.20 | $15.38_{\pm11.48}$ | $15.01_{\pm4.15}$ | $28.09_{\pm9.02}$ | $43.75_{\pm8.46}$ | $34.06_{\pm21.60}$ | $52.11_{\pm11.30}$ | $15.99_{\pm4.04}$ | $22.15_{\pm2.70}$ | $19.99_{\pm0.16}$ |
| | 5 | 0.01 | $67.71_{\pm2.51}$ | $22.39_{\pm6.18}$ | $66.05_{\pm9.07}$ | $64.69_{\pm4.15}$ | $60.47_{\pm2.44}$ | $68.46_{\pm4.46}$ | $24.32_{\pm3.60}$ | $25.61_{\pm2.11}$ | $20.69_{\pm1.76}$ |
| | | 0.05 | $17.55_{\pm8.71}$ | $17.27_{\pm1.84}$ | $42.76_{\pm5.67}$ | $63.75_{\pm4.65}$ | $59.38_{\pm4.79}$ | $67.77_{\pm4.82}$ | $20.41_{\pm2.47}$ | $20.53_{\pm5.54}$ | $19.73_{\pm1.13}$ |
| | | 0.20 | $12.98_{\pm2.29}$ | $18.12_{\pm1.64}$ | $32.68_{\pm9.75}$ | $59.69_{\pm4.41}$ | $43.59_{\pm15.76}$ | $59.54_{\pm8.94}$ | $20.03_{\pm2.67}$ | $22.14_{\pm2.80}$ | $20.63_{\pm2.12}$ |
| GRACE | 1 | 0.01 | $39.71_{\pm13.16}$ | $16.09_{\pm3.60}$ | $65.10_{\pm6.39}$ | $47.34_{\pm11.48}$ | $27.97_{\pm12.94}$ | $39.20_{\pm12.04}$ | $27.28_{\pm2.02}$ | $24.12_{\pm3.53}$ | $23.98_{\pm2.95}$ |
| | | 0.05 | $20.92_{\pm6.90}$ | $16.51_{\pm2.27}$ | $46.31_{\pm14.86}$ | $49.06_{\pm12.40}$ | $27.34_{\pm13.95}$ | $38.40_{\pm12.37}$ | $23.64_{\pm5.95}$ | $22.62_{\pm1.95}$ | $19.62_{\pm1.70}$ |
| | | 0.20 | $13.87_{\pm6.26}$ | $15.01_{\pm2.60}$ | $32.15_{\pm9.30}$ | $31.72_{\pm11.89}$ | $40.00_{\pm13.23}$ | $34.17_{\pm10.49}$ | $19.99_{\pm5.26}$ | $18.44_{\pm4.36}$ | $21.21_{\pm1.98}$ |
| | 3 | 0.01 | $61.75_{\pm6.43}$ | $51.58_{\pm7.30}$ | $66.66_{\pm6.50}$ | $48.59_{\pm11.11}$ | $58.59_{\pm6.79}$ | $69.26_{\pm6.74}$ | $22.73_{\pm2.42}$ | $22.94_{\pm1.69}$ | $25.40_{\pm2.37}$ |
| | | 0.05 | $26.12_{\pm7.17}$ | $21.63_{\pm4.37}$ | $56.63_{\pm9.53}$ | $51.09_{\pm11.66}$ | $56.87_{\pm6.65}$ | $69.14_{\pm5.19}$ | $21.35_{\pm1.26}$ | $20.51_{\pm3.28}$ | $20.78_{\pm1.75}$ |
| | | 0.20 | $17.09_{\pm7.61}$ | $23.22_{\pm3.48}$ | $40.08_{\pm5.38}$ | $39.06_{\pm11.28}$ | $39.37_{\pm16.47}$ | $71.31_{\pm1.93}$ | $20.20_{\pm2.20}$ | $21.13_{\pm2.60}$ | $20.41_{\pm0.94}$ |
| | 5 | 0.01 | $69.56_{\pm2.07}$ | $57.83_{\pm2.23}$ | $73.53_{\pm3.16}$ | $64.69_{\pm1.81}$ | $61.72_{\pm5.01}$ | $65.60_{\pm3.92}$ | $29.97_{\pm2.75}$ | $25.04_{\pm1.06}$ | $24.63_{\pm1.39}$ |
| | | 0.05 | $20.98_{\pm3.35}$ | $19.95_{\pm6.61}$ | $60.38_{\pm10.46}$ | $63.75_{\pm1.82}$ | $61.61_{\pm4.32}$ | $66.29_{\pm5.21}$ | $23.37_{\pm3.42}$ | $18.96_{\pm3.31}$ | $21.91_{\pm2.34}$ |
| | | 0.20 | $8.84_{\pm3.80}$ | $17.05_{\pm1.69}$ | $35.34_{\pm9.91}$ | $60.31_{\pm2.55}$ | $51.25_{\pm11.72}$ | $60.69_{\pm8.42}$ | $17.47_{\pm2.03}$ | $18.62_{\pm3.24}$ | $19.61_{\pm0.24}$ |
| GraphMAE | 1 | 0.01 | $40.25_{\pm8.40}$ | $30.03_{\pm7.41}$ | $56.54_{\pm8.91}$ | $50.94_{\pm8.48}$ | $42.34_{\pm14.66}$ | $64.80_{\pm13.22}$ | $22.59_{\pm1.50}$ | $21.85_{\pm0.82}$ | $22.42_{\pm2.31}$ |
| | | 0.05 | $10.58_{\pm1.47}$ | $18.42_{\pm1.29}$ | $40.30_{\pm10.47}$ | $50.00_{\pm11.42}$ | $45.78_{\pm16.96}$ | $63.54_{\pm13.43}$ | $19.49_{\pm1.18}$ | $23.89_{\pm1.29}$ | $20.16_{\pm0.46}$ |
| | | 0.20 | $10.59_{\pm2.95}$ | $18.31_{\pm1.75}$ | $32.69_{\pm6.86}$ | $39.69_{\pm16.35}$ | $48.13_{\pm13.11}$ | $57.03_{\pm13.33}$ | $18.97_{\pm1.51}$ | $21.35_{\pm2.35}$ | $20.42_{\pm0.47}$ |
| | 3 | 0.01 | $67.07_{\pm7.47}$ | $43.01_{\pm4.05}$ | $65.04_{\pm3.90}$ | $60.31_{\pm3.68}$ | $35.31_{\pm9.90}$ | $57.94_{\pm12.90}$ | $25.45_{\pm0.81}$ | $25.52_{\pm1.82}$ | $19.97_{\pm0.86}$ |
| | | 0.05 | $34.90_{\pm14.43}$ | $16.63_{\pm2.18}$ | $30.69_{\pm6.61}$ | $59.53_{\pm3.87}$ | $33.91_{\pm10.57}$ | $56.43_{\pm12.05}$ | $19.41_{\pm1.04}$ | $25.70_{\pm3.06}$ | $19.99_{\pm0.55}$ |
| | | 0.20 | $14.50_{\pm4.97}$ | $18.24_{\pm0.72}$ | $28.54_{\pm9.41}$ | $55.00_{\pm8.85}$ | $33.44_{\pm18.88}$ | $58.40_{\pm11.46}$ | $19.45_{\pm2.17}$ | $20.77_{\pm4.98}$ | $20.13_{\pm0.64}$ |
| | 5 | 0.01 | $64.44_{\pm3.70}$ | $16.86_{\pm4.51}$ | $71.46_{\pm3.32}$ | $67.06_{\pm1.17}$ | $66.56_{\pm9.12}$ | $75.09_{\pm4.83}$ | $25.01_{\pm1.85}$ | $24.49_{\pm1.88}$ | $23.31_{\pm0.74}$ |
| | | 0.05 | $34.30_{\pm11.05}$ | $20.61_{\pm6.32}$ | $49.20_{\pm9.18}$ | $68.44_{\pm1.45}$ | $65.78_{\pm8.43}$ | $74.29_{\pm4.28}$ | $22.34_{\pm2.96}$ | $20.11_{\pm5.47}$ | $19.92_{\pm1.20}$ |
| | | 0.20 | $14.87_{\pm5.61}$ | $17.05_{\pm1.69}$ | $37.75_{\pm4.10}$ | $61.25_{\pm2.19}$ | $54.22_{\pm11.96}$ | $74.74_{\pm3.01}$ | $19.83_{\pm1.78}$ | $17.46_{\pm5.73}$ | $19.84_{\pm0.61}$ |

## B.7 Computational Overhead Comparison

we provide the 1-shot time and space table for various baselines of DGI pretrained models, are shown in the Table 9. In the table, UniPrompt demonstrates efficient performance in both time and GPU costs. In terms of inference time, our approach is comparable to all GPL baselines. While our use of $k$NN slightly increases GPU memory usage, it remains within an acceptable range. This indicates that our method is lightweight and can be quickly deployed on various datasets, showcasing its efficiency.

## B.8 Robustness Analysis

Since our prompt topology is built on node features, it is sensitive to feature noise, which can lead to distorted graph structures. When both features and topology are misaligned with the pre-trained model, our method faces challenges in solving this problem. The 1-shot learning results under varying levels of Gaussian noise are summarized in the Table 10, where we observe that a 0.01 noise level shows minor augmentation, maintaining accuracy in some datasets (e.g. *PubMed*, *Wisconsin* and *Actor*). However, 0.05 noise begins to impact performance, and 0.20 noise significantly degrades accuracy across most datasets, with an average accuracy drop of over 30%.

## B.9 Hyperparameter Settings

We conduct extensive experiments to explore the impact of various hyperparameters on the performance of our model, as shown in Table 11, ensuring that our approach achieves robust and consistent results across diverse settings.

## C Related Works

**Graph Pretraining.** Graph pretraining has emerged as a powerful paradigm for learning generalizable and transferable representations from large-scale unlabeled graph data, aiming to mitigate the dependency on labeled data in downstream tasks. Unlike traditional supervised methods that require extensive manual annotations [3, 4, 48], graph pretraining leverages self-supervised strategies to capture structural and semantic patterns in graphs. Graph Self-Supervised Learning (GSSL) [49] currently has attracted widespread attention in the graph community, which mainly designs self-supervised objective functions to train the model based on maximizing Mutual Information (MI). As a classic paradigm, DGI [28] maximizes mutual information between node representations and

Table 11: Hyperparameter settings of `UniPrompt` for 1-shot, 3-shot, and 5-shot scenarios across different pretrained models

| Pretrain | Dataset | 1-shot | | | | 3-shot | | | | 5-shot | | | |
|---|---|---|---|---|---|---|---|---|---|---|---|---|---|
| | | up_lr | down_lr | $k$ | $\tau$ | up_lr | down_lr | $k$ | $\tau$ | up_lr | down_lr | $k$ | $\tau$ |
| DGI | Cora | 0.001 | 0.05 | 50 | 0.99999 | 0.0005 | 0.05 | 10 | 0.9999 | 0.0001 | 0.05 | 10 | 0.99999 |
| | CiteSeer | 0.0005 | 0.05 | 50 | 0.9999 | 0.0005 | 0.05 | 10 | 0.9999 | 0.0001 | 0.05 | 10 | 0.9999 |
| | PubMed | 0.0005 | 0.001 | 1 | 0.9999 | 0.0001 | 0.05 | 50 | 0.9999 | 0.0005 | 0.05 | 10 | 0.99999 |
| | Cornell | 0.001 | 0.0005 | 50 | 0.99 | 0.001 | 0.01 | 50 | 0.9999 | 0.00005 | 0.001 | 50 | 0.9999 |
| | Texas | 0.00001 | 0.0001 | 50 | 0.999 | 0.0001 | 0.00005 | 50 | 0.9999 | 0.00001 | 0.0001 | 50 | 0.9999 |
| | Wisconsin | 0.0001 | 0.001 | 50 | 0.999 | 0.005 | 0.0001 | 50 | 0.9999 | 0.00001 | 0.0001 | 50 | 0.9999 |
| | Chameleon | 0.00005 | 0.001 | 10 | 0.9999 | 0.00001 | 0.05 | 10 | 0.999 | 0.00001 | 0.05 | 10 | 0.999 |
| | Actor | 0.001 | 0.01 | 50 | 0.999 | 0.00001 | 0.01 | 50 | 0.9999 | 0.0005 | 0.005 | 50 | 0.9999 |
| | Squirrel | 0.00005 | 0.005 | 50 | 0.99999 | 0.0005 | 0.01 | 50 | 0.99 | 0.0001 | 0.0001 | 50 | 0.9999 |
| GRACE | Cora | 0.001 | 0.005 | 50 | 0.9999 | 0.001 | 0.05 | 50 | 0.9999 | 0.001 | 0.05 | 50 | 0.9999 |
| | CiteSeer | 0.005 | 0.001 | 50 | 0.9999 | 0.00001 | 0.05 | 50 | 0.9999 | 0.00001 | 0.05 | 50 | 0.9999 |
| | PubMed | 0.01 | 0.05 | 1 | 0.9999 | 0.01 | 0.05 | 1 | 0.9999 | 0.01 | 0.0001 | 1 | 0.9999 |
| | Cornell | 0.0001 | 0.0005 | 50 | 0.99 | 0.00001 | 0.0001 | 50 | 0.9999 | 0.00001 | 0.0005 | 50 | 0.9999 |
| | Texas | 0.0001 | 0.00005 | 50 | 0.9999 | 0.00005 | 0.0001 | 50 | 0.9999 | 0.00005 | 0.0005 | 50 | 0.9999 |
| | Wisconsin | 0.0001 | 0.01 | 50 | 0.999 | 0.0001 | 0.0005 | 50 | 0.999 | 0.0001 | 0.0001 | 50 | 0.9999 |
| | Chameleon | 0.005 | 0.001 | 1 | 0.99999 | 0.001 | 0.001 | 50 | 0.9999 | 0.005 | 0.05 | 50 | 0.99999 |
| | Actor | 0.0005 | 0.01 | 50 | 0.9999 | 0.005 | 0.01 | 50 | 0.9999 | 0.005 | 0.05 | 50 | 0.9999 |
| | Squirrel | 0.01 | 0.05 | 50 | 0.9999 | 0.005 | 0.05 | 50 | 0.99999 | 0.005 | 0.05 | 50 | 0.99999 |
| GraphMAE | Cora | 0.0005 | 0.0005 | 50 | 0.9999 | 0.0005 | 0.05 | 1 | 0.99999 | 0.005 | 0.0005 | 1 | 0.9999 |
| | CiteSeer | 0.001 | 0.0001 | 1 | 0.99999 | 0.001 | 0.05 | 50 | 0.9999 | 0.001 | 0.05 | 10 | 0.9999 |
| | PubMed | 0.005 | 0.01 | 1 | 0.999 | 0.0001 | 0.05 | 10 | 0.9999 | 0.0001 | 0.05 | 1 | 0.9999 |
| | Cornell | 0.00005 | 0.05 | 50 | 0.9999 | 0.00005 | 0.005 | 50 | 0.9999 | 0.00005 | 0.0005 | 50 | 0.9999 |
| | Texas | 0.00001 | 0.0005 | 50 | 0.9999 | 0.00005 | 0.0005 | 50 | 0.9999 | 0.00005 | 0.0005 | 50 | 0.9999 |
| | Wisconsin | 0.00005 | 0.01 | 50 | 0.9999 | 0.00001 | 0.00005 | 50 | 0.9999 | 0.00001 | 0.0005 | 50 | 0.9999 |
| | Chameleon | 0.00001 | 0.005 | 50 | 0.99999 | 0.001 | 0.001 | 50 | 0.9999 | 0.001 | 0.05 | 50 | 0.9999 |
| | Actor | 0.005 | 0.05 | 50 | 0.9999 | 0.001 | 0.05 | 50 | 0.9999 | 0.01 | 0.05 | 50 | 0.9999 |
| | Squirrel | 0.005 | 0.05 | 50 | 0.9999 | 0.001 | 0.05 | 5 | 0.99999 | 0.005 | 0.0001 | 50 | 0.9999 |

Table 12: Settings and code links of various baseline methods.

| Methods | Source Code |
|---|---|
| $k$-Shot Sampling | ProG/blob/main/prompt_graph/tasker/node_task.py |
| Dataset Split | ProG/blob/main/prompt_graph/data/load4data.py |
| Evaluation | ProG/blob/main/prompt_graph/evaluation/AllInOneEva.py |
| DGI | https://github.com/PetarV-/DGI |
| GRACE | https://github.com/CRIPAC-DIG/GRACE |
| GraphMAE | https://github.com/THUDM/GraphMAE/tree/pyg |
| GPPT | https://github.com/MingChen-Sun/GPPT |
| GraphPrompt | https://github.com/Starlien95/GraphPrompt |
| GPF/GPF+ | https://github.com/zjunet/GPF |
| All-in-one | https://github.com/sheldonresearch/ProG |
| EdgePrompt/EdgePrompt+ | https://github.com/xbfu/EdgePrompt |

the summary of the graph. GGD [50] further explores the DGI, summarizing it as a group discrimination task, greatly reducing the computational time overhead. MVGRL [51] introduces graph diffusion to generate different scale subgraphs to improve the pipeline of DGI. GRACE [29] uses InfoNCE [45, 46] to optimize by maximizing the similarity of two augmented nodes generated by the same node and minimizing the similarity of other nodes to train the model. GCA [47] improve the augmentation strategy by defining the importance of different nodes and edges to preserve semantic information. GraphCL [37] and JOAO [52] use this paradigm to global-global graph representation contrast. AD-GCL [53] has similar idea, which designs a learnable augmentation strategy, and trains the model by maximizing mutual information between node representation and augmented graph. Some works [54, 55, 56, 57, 58, 59, 60, 61] focus on design effective sampling strategy, and other works [62, 63] further introduce additional knowledge into GCL. AFGRL [64], SimGRACE [65], NeCo [66] and AFGCL [67] propose augmentation free paradigm to optimize sampling. PolyGCL [68] and S3GCL [69] focus on designing polynomials with learnable filters to generate different spectral contrastive views. GraphMAE [30], as the method of masked graph autoencoders, utilizes randomly mask mechanism from the input with the graph autoencoder [70] architecture to reconstruct the node features or structures. GraphMAE2 [71] proposes multiview

random re-mask decoding, the node representations are randomly re-masked multiple times, to introduce randomness into feature reconstruction. BGRL [31] adopts BYOL [72], which trains the online encoder by predicting the target encoder to generate efficient node representations. This backbone is followed by some recent works, such as SGRL [73] and SGCL [32].

**Graph Prompt Learning.** Graph prompt learning aims to address the gap between pretrained models and downstream tasks by introducing tunable components into the inputs, model parameters, or outputs of pretrained models. This approach facilitates the alignment of the pretraining domain with the target domain, thereby improving performance in downstream tasks, particularly in few-shot fine-tuning scenarios. GPPT [6] introduces structure tokens and task tokens, transforming the node classification task into a form consistent with link prediction. GPF/GPF-plus [7], from the perspective of the feature space, inject global and specific prompt vectors into nodes, bringing the prompt-tuning paradigm into graph representation learning. SUPT [74] extends GPF-plus to the subgraph level, modifying the attention mechanism to a GCN-like aggregation that incorporates neighborhood information. IA-GPL [75] further advanced this by introducing an instance-aware mechanism that maps node representations to a prompt space, and quantizes them into a codebook using Vector Quantization (VQ) [76]. The resulting quantized prompts are then combined with node features as input for the pre-trained model. RELIEF [77] employs reinforcement learning to select a small, efficient set of nodes for prompt generation, creating node-specific prompts to avoid the potential interference of applying prompts to all nodes. All-in-one [14] unifies the "pretraining-prompt" paradigm by converting node and edge tasks into graph-level tasks by building an induced subgraph. It then adds learnable prompts to the node features in a weighted manner to construct a prompt graph. By combining meta-learning with this process, the prompts can adapt to multiple downstream tasks. HeterGP [78] extends this paradigm by considering heterophilic scenarios. GraphPrompt [24] unifies node-level and graph-level tasks into subgraph similarity computation and incorporates a learnable prompt vector into the readout layer of GNNs, enabling the model to adapt flexibly to various downstream tasks. Building on this, GraphPrompt+ [15] generalizes pre-training tasks to arbitrary contrastive learning tasks and introduces prompt vectors into each GNN layer, thereby leveraging hierarchical knowledge. MultiGPrompt [26] further adopts multi-task pre-training to learn more comprehensive and multi-level representations. PRODIGY [79] introduces task graphs to unify pretraining and downstream tasks via in-context learning. It avoids parameter tuning by reformulating tasks as link prediction between data and label tokens. GraphControl [22] aligns cross-domain graphs via conditional prompts inspired by ControlNet [80], enabling semantic consistency in transfer learning. Moreover, some works are designed for considering heterogeneous graphs. HGPrompt [81], based on GraphPrompt, decomposes a heterogeneous graph into multiple homogeneous subgraphs and introduces feature prompts and heterogeneity prompts, thereby proposing the prompt framework applicable to both homogeneous and heterogeneous graphs. HetGPT [82] further introduces virtual class prompts and heterogeneous feature prompts, and adopts a multi-view neighborhood aggregation mechanism to effectively model the complexity of heterogeneous neighborhood structures, constructing a general framework for heterogeneous graph prompting. HiGPT [83] targets more complex and dynamic heterogeneous graph scenarios by employing context-parameterized heterogeneity projectors and LLMs to generate node embeddings, while leveraging instruction tuning in downstream tasks to enhance generalization ability. Although various graph prompt strategies have advanced the field, there remains no unifed understanding of how these prompts interact with pretrained models, which is the problem our work tries to explain and solve.

**Prompt Techniques in Graph Foundation Models.** Due to its simple and efficient design, prompt techniques is widely used in some graph foundation models and LLM+GNN paradigms. GCOPE [38] proposes the concept of a "Coordinator", which introduces a set of virtual nodes to bridge different datasets, enabling the model to learn knowledge across multiple domains and transfer it to a wide range of downstream tasks. OFA [16] and GOFA [84] introduce LLMs into graph learning. Specifically, OFA [16] first transforms all graph data into text-attributed graphs (TAGs), and augments the NOI (node of interest) subgraph with NOI prompt nodes and class prompt nodes. The data are then processed sequentially by an LLM and a GNN to predict the category of the NOI. GOFA [84] interleaves GNN layers with LLM layers, which not only preserves the ability to learn graph structures but also equips the model with text generation capabilities, thereby enabling broader generalization to downstream tasks. GraphPrompter [85] projects TAGs into a semantic space to align them. A GNN then encodes the graph structure, and the resulting node embeddings are concatenated into prefix tokens. These tokens are prepended to the input text, enabling a frozen LLM to understand and reason about the graph data. ZeroG [12] uses LoRA [86] to fine-tune a pretrained Language Model (LM).

It creates a neighborhood-aware prompt that aggregates local topological information via message passing, helping the LM generate effective representations for zero-shot tasks. UniGraph [87] maps cross-domain graph features into a unified LLM semantic space. It trains a cascaded LM-GNN encoder using a masked language modeling task. For downstream tasks, it uses in-context learning for few-shot transfer and combines LoRA with fine-tuning for zero-shot transfer. LLaGA [88] transforms graph structures into node sequences using either a neighbor-based or hop-count based approach. The resulting sequences, which are rich in structural information, are combined with prompt tokens and fed into a frozen LLM for various downstream tasks. GraphGPT [89] uses a two-phase prompt tuning to enhance LLM understanding of graph structures. The first phase uses self-supervised prompt tuning with a graph matching task to learn a projector. The second phase involves task-specific fine-tuning to customize the reasoning of LLMs for different downstream tasks. The GFT [90] framework defines a computation tree as a token, and uses VQ to maintain a token vocabulary during pretraining. For downstream prompting, any task can be re-framed as the classification of these computation tree tokens. GIT [91] adopts similar concepts to GFT, but adds theoretical proofs to demonstrate the stability, transferability, and generalization of the task tree.

## D  Limitations

Despite the excellent results achieved by our proposed `UniPrompt` method, several limitations should be considered:

- **Limited Integration with LLMs**: Our proposed method currently focuses on adapting traditional graph pretrained models and does not explore the integration of Large Language Models (LLMs) as encoders. This is a notable limitation, given the increasing prominence of LLMs in generating powerful, semantically rich node representations from textual attributes. The full potential of `UniPrompt` in semantically driven graph tasks and its applicability to emerging paradigms like zero-shot graph learning remain unexplored.

- **Hyperparameter Dependency**: The effectiveness of `UniPrompt` hinges on two key hyperparameters: the temperature coefficient, $\tau$, which balances the original and prompt topologies, and the number of neighbors, $k$, in the $k$NN graph. The hyperparameter analysis reveals that the optimal settings for these parameters vary significantly across datasets of different types (e.g., homophilic vs. heterophilic) and scales. This necessitates careful tuning when applying the method to new datasets, which adds to its practical complexity.

- **Limited Task Coverage**: The current evaluation is exclusively focused on node classification, particularly in few-shot settings. Whether `UniPrompt` can be effectively generalized to other important graph learning tasks, such as graph classification, link prediction, or community detection, remains unverified. These tasks have different requirements for global structural information or edge-level relationships, for which the current node centric topological prompt may not be optimal.

## E  Broader Impacts

The introduction of `UniPrompt` represents a significant advancement in the field of graph prompt learning. The broader impact of this work includes:

- **Promoting the Development of Graph Foundation Models**: By theoretically dissecting the underlying mechanisms of different prompting strategies, this research proposes a clearer design paradigm: graph prompt learning should focus on unleashing the capability of pretrained models, and the classifier adapts to downstream scenarios. This perspective offers theoretical guidance for building more powerful and versatile Graph Foundation Models (GFMs), helping to steer the field toward a more unified and efficient direction.

- **Further Integration with LLMs**: The principles of `UniPrompt` can be powerfully combined with Large Language Models (LLMs), which have become de facto foundation models for language. In this paradigm, an LLM could serve as a powerful feature encoder for text-attributed graphs, and `UniPrompt`'s adaptive topology would then refine the graph structure to best suit the rich semantic representations provided by the LLM. This integration would test the universality of our approach, demonstrating how a learnable, input-level prompt can

effectively guide a powerful, but non-graph-native, pre-trained model to reason over graph structures.

- **Extension to Multiple Tasks**: While the current work focuses on node classification, the core idea of learning an adaptive topology is task-agnostic and holds significant promise for other fundamental graph tasks. For graph classification, the learned prompt graph could highlight key subgraphs or motifs crucial for determining a graph's overall label. For link prediction, the adaptive topology could help the model capture higher-order structural patterns predictive of missing edges. Extending `UniPrompt` to these diverse tasks would be a crucial step in validating its effectiveness as a more universal adaptation method for graph models.

- **Enhancing Model Robustness and Generalization**: The method enhances the generalization capability of pre-trained models by learning a topology that is adaptive to the downstream task, especially when handling distribution shifts (e.g., from homophily to heterophily). This concept can inspire further research into improving cross-domain adaptation and Out-of-Distribution (OOD) generalization, which is vital for building reliable and stable AI systems that can operate in diverse, real-world environments.

