# OpenReview forum: "One Prompt Fits All: Universal Graph Adaptation for Pretrained Models"
_NeurIPS.cc/2025/Conference — NeurIPS 2025 poster_

### Official Review · Reviewer_oJWJ · 2025-06-27

**Clarity:** 2
**Significance:** 2
**Originality:** 2
**Rating:** 4
**Confidence:** 3

**Summary:**

This paper introduces a graph prompt learning (GPL) method, UniPrompt for unleashing the full pretrained graph models by leveraging input-level and layer-wise prompts while preserving the structure of the input graph. It identifies key limitations in existing GPL methods, such as a lack of consensus on underlying mechanisms and limited scenario adaptability. The authors conduct a theoretical analysis, showing that representation-level prompts are equivalent to fine-tuning a simple classifier, and argue that effective GPL should focus on enabling pretrained models to understand input data. Extensive experiments demonstrate the effectiveness of UniPrompt across multiple datasets in both in-domain and cross-domain scenarios.

**Questions:**

1.How well would UniPrompt perform on real-world large-scale graphs, such as social networks or citation networks, where the kNN initialization and bootstrapping strategy might become computationally intensive?
2.Why was kNN chosen as the basis for prompt graph initialization? Have other initialization strategies, such as random or learned topologies, been explored?
3.The paper demonstrates that UniPrompt performs well across different pretrained models, but does the method rely on specific pretrained graph embeddings (e.g., DGI or GraphMAE), or is it agnostic to the pretraining approach?
4.How sensitive is UniPrompt to the choice of hyperparameters like the temperature coefficient (τ) and the number of neighbors (k)? Could these parameters severely impact performance in certain scenarios?

**Ethical Concerns:**

["NO or VERY MINOR ethics concerns only"]

**Final Justification:**

The authors have provided a detailed and satisfactory response to my concerns, so I am increasing my rating.

**Limitations:**

YES

**Quality:**

2

**Strengths And Weaknesses:**

Strengths:
1.The paper offers a perspective by emphasizing the importance of unleashing pretrained model capabilities rather than adapting them to downstream tasks.
2.The authors evaluate UniPrompt on a wide variety of datasets, including homophilic and heterophilic graphs, and under different few-shot settings, providing robust empirical evidence.

Weaknesses:
1.While the perspective on representation-level prompts is interesting, the core methodology (e.g., kNN-based topology prompts, bootstrapping strategies) appears incremental and builds heavily on existing ideas in graph structure learning (GSL) and self-supervised learning.
2.The theoretical results focus primarily on representation-level prompts but do not provide a rigorous analysis of input-level and layer-wise prompts, which are central to the proposed UniPrompt method.
3.While the experiments are comprehensive, there is a lack of detailed ablation studies to isolate the contributions of individual components of UniPrompt, such as the kNN-based prompt initialization or the bootstrapping strategy.
4.The paper highlights the strengths of UniPrompt but does not sufficiently discuss its limitations or scenarios where it may fail (e.g., datasets with extremely high heterophily or noisy features).

---

> ### Author Rebuttal · Authors · 2025-07-30
>
> Thank you for your constructive feedback; we appreciate your insights and have addressed your comments below.
> ### **To W1 & Q2:**
>
> We thank the reviewer for the thoughtful comments.
>
> While techniques like kNN and bootstrapping have been used in GSL and SSL, our work is different in how we apply them to graph prompt learning, with the goal of unleashing the capabilities of pretrained GNN models. The kNN algorithm is consistent with the homophily assumption underlying widely adopted pre-trained GNN models. By constructing a similarity-based graph, kNN enables the input data to better align with these models, allowing information from similar nodes to be aggregated, thereby enabling more effective adaptation. This design aligns well with our core perspective: graph prompt learning should focus on unleashing the capability of pretrained models. Compared to feature-based or connection-based prompt injection methods, kNN provides a clear and structure-based method that help the pretrained GNN models work more effectively.
>
> - For random topology: Randomly linked graphs may lack meaningful structure, hindering the pretrained model's ability to capture node similarity or semantic relationships. This severely impacts the performance of downstream tasks, with an average accuracy drop of over 25%. Detailed experiments are provided in the response to weakness 3.
>
> - For learnable topology: While the idea of end-to-end learnable topology is appealing, it typically requires complex design and greater computational overhead to ensure the effectiveness of the learned structure. We agree that this represents a promising direction for future research, and we thank you for highlighting this point.
>
>
> ### **To W2:**
>
> We categorize existing graph prompt methods into three main types according to their mechanisms: input-level, layer-wise, and representation-level. We want to analyze these various approaches to gain a deeper understanding of the underlying prompt mechanism.
>
> Regarding representation-level methods, our analysis demonstrates that they are equivalent to a classifier. This suggests that their mechanism contradicts the core advantages of prompting and fails to leverage the benefits of prompts, as similar results can be achieved with a classifier alone. As for layer-wise methods, due to their inherent design complexity and strong reliance on the internal representations of pre-trained models, they are not suitable for black-box pre-trained GNNs. Therefore, we do not consider these methods in our work.
>
> In contrast, input-level methods avoid the limitations and preserve the core advantages of prompting without requiring access to model internals. Thus, they are the most promising among the three categories for our setting. We thus design an input-level prompt mechanism. Through this process, we get our core perspective: "Graph prompt learning should focus on unleashing the capability of pretrained models, and the classifier adapts to downstream scenarios.", which clearly defines the distinct roles and mechanisms of the two crucial downstream components: the prompt and the classifier.
>
> ### **To W3:**
>
> We appreciate this suggestion. Since both components are essential and cannot be simply removed, we conduct replacement experiments: (1) replacing kNN with random topology, (2) replacing bootstrapp with simple addition of original and prompt graph, (3) discarding original graph totally. 1-shot results as shown in the table below:
>
> | DGI-pretrained | Cora       | CiteSeer   | PubMed     | Cornell     | Texas       | Wisconsin  | Chameleon  | Actor      | Squirrel   |
> | --------------------- | ---------- | ---------- | ---------- | ----------- | ----------- | ---------- | ---------- | ---------- | ---------- |
> | Random_Topo           | 44.06±8.80 | 36.69±9.18 | 52.85±4.00 | 27.81±11.11 | 18.59±3.78  | 35.43±4.06 | 20.79±2.27 | 20.73±2.32 | 22.33±1.58 |
> | Simple_Add          | 24.54±4.95 | 26.27±3.37 | 43.80±9.69 | 51.72±16.49 | 38.28±13.17 | 55.89±4.91 | 23.34±3.90 | 23.47±2.09 | 23.56±1.20 |
> | Discard_Topo          | 28.17±6.44 | 30.15±6.88 | 37.33±6.77 | 52.03±16.25 | 46.72±13.46 | 52.63±8.35 | 23.20±4.03 | 24.95±3.74 | 22.31±1.34 |
>
> From the table, we can find that Random_Topo maintains some of effectiveness on homophilic datasets while showing reduced performance on heterophilic ones. Conversely, for Simple Addition and Discarding Topology, heterophilic datasets still retain some performance. However, performance on homophilic datasets drops significantly, as their original structure is crucial for classification. This table will be included in the revised versions.
>
> ### **To W4:**
> We appreciate you raising this point. Since our prompt topology is built on node features, it is sensitive to feature noise, which can lead to distorted graph structures. When both features and topology are misaligned with the pre-trained model, our method faces challenges in solving this problem. The 1-shot learning results under varying levels of Gaussian noise are summarized in the following table:
>
> | DGI-pretrained | Cora        | CiteSeer    | PubMed     | Cornell     | Texas       | Wisconsin   | Chameleon  | Actor      | Squirrel   |
> | --------------------- | ----------- | ----------- | ---------- | ----------- | ----------- | ----------- | ---------- | ---------- | ---------- |
> | Noisy-0.01            | 44.42±10.49 | 32.39±12.85 | 61.00±5.31 | 50.62±14.07 | 46.72±12.88 | 62.51±10.50 | 20.80±2.43 | 26.96±4.35 | 22.50±2.07 |
> | Noisy-0.05            | 23.23±9.21  | 19.77±1.66  | 40.01±0.98 | 49.22±12.02 | 38.75±13.32 | 61.03±9.79  | 20.72±0.76 | 24.67±2.96 | 20.41±0.74 |
> | Noisy-0.2             | 27.82±5.26  | 15.73±5.90  | 39.46±0.37 | 28.59±5.96  | 33.91±10.64 | 42.86±16.34 | 20.08±2.77 | 22.15±2.70 | 20.21±0.27 |
>
> We observe that a 0.01 noise level shows minor augmentation, maintaining accuracy in some datasets (e.g. PubMed, Wisconsin and Actor). However, 0.05 noise begins to impact performance, and 0.20 noise significantly degrades accuracy across most datasets, with an average accuracy drop of over 30%. This analysis will be incorporated into future versions.
>
> For the extreme heterophily cases, our method can address these situations. Our experiments demonstrate that our approach works effectively on different heterophilic datasets. Conversely, when the alignment between upstream and downstream data is already relatively high (e.g. both training and testing use homophilic datasets), our improvement is less pronounced. This discussion will be incorporated into future versions.
>
> ### **To Q1:**
>
> Thank you for the suggestion. We have additionally run experiments on the large-scale heterophilic dataset Arxiv-year (169,343 nodes and 1,166,243 edges) as a supplement. Here, we use a simplified kNN by randomly sampling 1000 nodes, then connecting each node to its top-k most similar sampled nodes. We test three pretrain strategies under 5-shot setting, comparing with fine-tuning. The accuracy and computational cost are shown in the table below.
>
> | 5-shot               | Arxiv-Year (acc) | Preprocessing Time (s) | Training Time (s/per_epoch) |
> | -------------------- | ---------------- | :--------------------- | --------------------------- |
> | Fine-tune (DGI)      | 28.27±5.99       | -                      | 0.0138                      |
> | Ours (DGI)           | 32.48±6.37       | 1.25                   | 0.0224                      |
> | Fine-tune (GRACE)    | 24.60±1.04       | -                      | 0.0205                      |
> | Ours (GRACE)         | 25.17±2.83       | 1.26                   | 0.0320                      |
> | Fine-tune (GraphMAE) | 23.24±1.58       | -                      | 0.0427                      |
> | Ours (GraphMAE)      | 24.25±5.43       | 1.32                   | 0.0618                      |
>
> As shown in the table, our method incurs minimal preprocessing time and only a slight increase in training time per epoch, with small epoch counts (typically less than 500). This demonstrates that our approach is scalable to large graphs. This table will be included in the future versions.
>
> ### **To Q3:**
>
> Thank you for your thoughtful consideration. Our method is designed to be agnostic to the choice of pretrained GNN. To demonstrate this, we evaluate UniPrompt on diverse pretrained backbones, including DGI, GRACE, GraphMAE and the cross-domain model FUG. The consistent performance across these settings supports the generality of our approach.
>
> That said, we acknowledge a limitation: most pretrained models we use are trained on homophilic datasets, and our similarity-based prompting mechanism naturally aligns with such inductive biases. If pretrained GNN models or foundation models were trained exclusively on heterophilic graphs, the effectiveness of our current strategy may vary. However, this does not conflict with current practice, nearly all pretrained GNN models and foundation models are trained on homophilic benchmarks. We intend to adapt our work to heterophilic settings as future work.
>
> ### **To Q4:**
> Thank you for your question, we provide hyperparameter analysis to $\tau$ and $k$ in Figure 4. In general, smaller $\tau$ perform better on heterophilic graphs while larger $\tau$ work better on homophilic graphs. This is consistent with the characteristics of real-world datasets, where adding more homophilic information to heterophilic datasets can further leverage topological information to improve performance. A similar principle applies to $k$, smaller $k$ works well when pretrained models are trained on homophilic graphs, and larger $k$ can enhance the intervention of the prompt graph in heterophilic graphs.
>
> While these hyperparameters do influence performance, they are not prohibitively sensitive. In practice, prior knowledge of graph homophily can guide the selection of $\tau$ and $k$, reducing the tuning burden. Future work will explore learnable or adaptive $\tau$ and $k$ to automate the process further.

---

> > ### Author Response · Authors · 2025-08-05
> > **A gentle follow-up.**
> >
> > Thank you again for your valuable feedback on our submission. We have submitted our rebuttal and are looking forward to hearing your further thoughts on the changes. Please let us know if there is anything else we can provide.

---

### Official Review · Reviewer_5pxd · 2025-06-30

**Clarity:** 3
**Significance:** 3
**Originality:** 4
**Rating:** 5
**Confidence:** 4

**Summary:**

This work introduces a novel graph prompt tuning method, which is to address the lack of mechanism analysis and limited scenario adaptability. The authors theoretically prove that "representation-level" prompts are equivalent to linear probe, and they finds that prompt tuning should focus more on unleashing pre-trained model capabilities. Specifically, the method conducts the edge prompt using a kNN graph, and employs a bootstrapping strategy to integrate the prompt graph with the original graph. Extensive experiments demonstrate that the proposed method achieves good performance over classic prompt tuning/foundation models baselines.

**Questions:**

The authors claim that "GPLs often experience performance instability or even negative optimization", and Figure 2 illustrates unstable convergence for GPPT. While the paper does not delve deeply into the fundamental reasons behind this "negative optimization" or instability. I think the authors need more detailed analysis for several reasons. For instance, are we seeing severe overfitting in the few-shot setting, gradient issues (e.g., vanishing or exploding gradients)? And if so, how does uniprompt effectively mitigate these problems?

**Ethical Concerns:**

["NO or VERY MINOR ethics concerns only"]

**Final Justification:**

It has addressed my concerns, and I will keep my score.

**Limitations:**

Yes

**Quality:**

3

**Strengths And Weaknesses:**

Strengths:

1. The analysis of existing graph prompt work is valuable. This paper rethinks and categorizes existing methods, yielding valuable conclusions that should influence the graph prompt community.

2. The theoretical analysis finds the equivalence between "representation-level" prompts and linear probe, and the authors propose that graph prompt tuning should focus more on unleashing the capabilities of pre-trained models. This perspective holds implications for the community of graph prompt tuning.

3. The experiments are comprehensive, achieving excellent results across various scenarios, and include extensions to some graph prompt tuning baselines.

4. The proposed method is simple yet powerful, and easy to follow.

Weaknesses:

1. Part of the Figure 1 is unclear. While I understand the authors' point, Figure 1a (input-level prompt) visually suggests adding a prompt to features, which doesn't fully capture the broader concept of "modifying the input graph" as described on Line 34. Figure 1 requires further improvement.

2. I think the paper's introduction to layer-wise prompt methods is insufficient. While these methods may lack extensive exploration in the graph domain, they have shown significant influence in other fields, such as [J1, C1]. I suggest the authors provide additional analysis or discussion of these methods.

3. The authors claim that "only fine-tuning a classifier can achieve or exceed the performance of existing GPLs.". However, this conclusion is only marginally demonstrated in the motivational experiments and isn't included in the main experiments. I think these linear probe results need to be fully added to the main table.

[J1] Visual Prompt Tuning, ECCV-22
[C1] MaPLe: Multi-modal Prompt Learning, CVPR-23

---

> ### Author Rebuttal · Authors · 2025-07-30
>
> Thank you for your support of our work! We appreciate your insights and have addressed your comments below.
>
> ### **To Weakness 1:**
>
> Thank you for pointing out this potential misunderstanding. The notation $\mathcal{G}$ in this figure is indeed unclear.
>
> As we stated in the preliminary section, under the general "pretrain, fine-tune" paradigm of graph representation learning, $ \mathcal{G} = (\mathbf{A}, \mathbf{X})$, which means it includes both feature and topological elements. Here, we generalize that input-level prompts can arbitrarily affect both components. We will revise Figure 1 to provide a more detailed and clear representation of this concept in the future version.
>
> ### **To Weakness 2:**
>
> Thank you for your valuable suggestions. We completely agree that the introduction to layer-wise prompt methods was insufficient in the initial draft. As you pointed out, while such methods have been less explored in the graph domain, they have already made significant impacts in fields such as vision and multimodal learning. We will include discussions of works from other domains such as VPT [J1] and MaPLe [C1] in future versions, and add detailed analyses between layer-wise prompts in graph domain versus vision/multimodal domains, including design methods, applicable scenarios, and performance differences across different data domains. we will include these discussion in the future version.
>
> ### **To Weakness 3:**
>
> Thank you for your suggestions. We provide the fully experiment of Fine-tune and Linear-probe here, as shown in the table below:
>
> | Method                     | Cora       | CiteSeer   | PubMed     | Cornell    | Texas       | Wisconsin   | Chameleon  | Actor      | Squirrel   |
> | -------------------------- | ---------- | ---------- | ---------- | ---------- | ----------- | ----------- | ---------- | ---------- | ---------- |
> | Fine_tuning (1-shot DGI)   | 50.22±9.28 | 42.58±8.87 | 53.90±8.30 | 35.23±8.84 | 37.50±13.57 | 33.91±10.56 | 24.42±3.19 | 21.36±3.28 | 22.27±4.10 |
> | Linear_probe  (1-shot DGI) | 49.77±9.74 | 43.16±7.60 | 55.76±9.43 | 34.26±8.60 | 36.21±13.77 | 28.71±9.38  | 23.64±2.17 | 21.33±2.62 | 22.82±4.10 |
> | Fine_tuning (1-shot GRACE)  | 48.59±9.20 | 46.16±6.30 | 57.97±7.55 | 34.18±10.18 | 31.52±13.08 | 32.23±8.96 | 26.22±2.73 | 20.81±2.86 | 21.16±2.57 |
> | Linear_probe (1-shot GRACE) | 46.22±7.92 | 46.10±6.32 | 57.87±7.60 | 34.92±9.74  | 34.84±15.65 | 31.66±8.18 | 24.27±3.84 | 20.53±3.11 | 20.81±1.82 |
> | Fine_tuning (1-shot GraphMAE)  | 45.92±9.67  | 36.47±8.35 | 54.29±9.52 | 35.82±11.30 | 37.07±14.08 | 33.54±10.16 | 22.08±3.19 | 20.85±1.68 | 21.32±2.65 |
> | Linear_probe (1-shot GraphMAE) | 50.13±12.06 | 48.08±6.96 | 58.61±8.34 | 32.27±11.28 | 38.32±13.61 | 28.40±8.67  | 23.02±2.08 | 20.56±2.91 | 21.05±1.87 |
> | Fine_tuning (3-shot DGI) | 65.09±5.73 | 60.32±4.05 | 64.81±6.80 | 41.84±6.52 | 38.75±9.34 | 41.34±7.57 | 28.66±3.99 | 22.61±2.36 | 23.02±3.54 |
> | Linear_probe (3-shot DGI) | 67.48±4.65 | 60.91±4.23 | 65.92±5.53 | 40.39±8.35 | 39.30±7.67 | 38.29±9.18 | 27.17±4.04 | 21.66±2.31 | 22.56±2.29 |
> | Fine_tuning (3-shot GRACE) | 63.99±5.69 | 60.48±4.52 | 62.03±6.10 | 42.42±7.46 | 39.73±6.45 | 40.34±5.37 | 29.73±4.02 | 21.70±2.14 | 23.77±2.23 |
> | Linear_probe (3-shot GRACE) | 63.68±5.99 | 60.35±3.99 | 65.71±5.87 | 41.60±7.19 | 39.53±6.62 | 41.46±5.28 | 29.02±4.60 | 21.55±1.38 | 21.62±3.47 |
> | Fine_tuning (3-shot GraphMAE) | 66.38±6.34 | 58.57±5.82 | 62.51±4.55 | 46.09±8.50 | 43.91±8.88 | 48.31±5.70 | 27.33±3.17 | 21.40±1.56 | 21.18±1.20 |
> | Linear_probe (3-shot GraphMAE) | 70.74±4.52 | 60.60±4.96 | 66.90±4.70 | 38.52±7.65 | 43.13±8.47 | 41.40±5.54 | 29.02±4.05 | 22.08±1.77 | 21.91±1.74 |
> | Fine_tuning (5-shot DGI) | 73.01±2.55 | 65.08±3.52 | 70.91±4.65 | 45.78±5.65 | 43.20±9.51 | 43.26±7.43 | 28.81±2.82 | 23.65±2.38 | 22.58±2.75 |
> | Linear_probe (5-shot DGI) | 72.39±2.01 | 65.11±2.62 | 70.32±4.19 | 45.23±6.87 | 42.81±8.09 | 41.66±5.68 | 28.80±2.67 | 22.55±2.40 | 23.53±1.70 |
> | Fine_tuning (5-shot GRACE) | 70.49±2.28 | 64.19±3.49 | 70.42±5.36 | 47.15±6.77 | 43.09±8.74 | 42.51±5.92 | 34.00±2.48 | 22.61±1.91 | 25.22±1.65 |
> | Linear_probe (5-shot GRACE) | 71.09±2.18 | 63.65±3.29 | 71.34±6.46 | 47.07±6.66 | 42.11±8.02 | 41.91±6.20 | 32.78±3.15 | 22.23±1.88 | 24.05±1.58 |
> | Fine_tuning (5-shot GraphMAE) | 73.85±2.87 | 64.59±4.32 | 72.83±3.21 | 58.24±4.44 | 47.62±6.96 | 50.29±6.59 | 28.78±2.47 | 21.22±4.17 | 22.38±1.12 |
> | Linear_probe (5-shot GraphMAE) | 75.78±2.38 | 66.17±2.72 | 70.08±4.82 | 43.71±5.71 | 45.00±7.98 | 41.11±7.54 | 31.31±3.63 | 22.51±2.23 | 22.25±1.75 |
>
> From the table, we observe that the performance of Linear-probe is comparable to Fine-tuning across most datasets. Notably, on homophilous datasets (e.g., Cornell, Wisconsin, and Texas), Linear-probe often outperforms Fine-tuning. In contrast, Fine-tuning tends to perform better on some heterophilous datasets.
>
> Comparing this table with Table 1 in the paper, we find that Linear-probe achieves competitive or even superior performance compared to multiple baselines, particularly for representation-level baselines (GPPT and GraphPrompt). The improvement is more pronounced on heterophilous datasets such as Cornell, Wisconsin, and Texas, while there are also slight gains on Chameleon, Actor, and Squirrel. We will include these results in the future version.
>
> ### **To Question:**
>
> Thank you for your question. We have discussed part of the potential scenarios in section 5, and we think that overfitting is a common issue in few-shot downstream settings. We attribute this to the use of inappropriate prompt mechanisms, which obscure the prompt's true function, leading to unstable convergence and even negative optimization during training. This phenomenon was part of our motivating experiments.
>
> To address this, first, we designed our prompt using kNN to provide the model with a stable structural prior. This transforms the prompt from discrete, random vectors into a stable structure built upon the semantic relationships between nodes. Second, we employed bootstrapping to further mitigate overfitting, ensuring that both the original topology and the prompt's topology are utilized effectively.
>
> Finally, our approach clarifies the roles of the two key downstream modules: the prompt and the classifier, allowing both to contribute effectively. This not only reduces the risk of overfitting but also ensures that the structural prompt smoothes the optimization landscape, leading to a more stable training process and alleviating potential gradient issues like exploding or vanishing gradients.

---

> > ### Comment · Reviewer_5pxd · 2025-08-01
> >
> > Thank you for the response. It has addressed my concerns, and I will keep my score.

---

> > > ### Author Response · Authors · 2025-08-05
> > > **Appreciation for the 5pxd's Feedback.**
> > >
> > > Thank you for your positive assessment. We are very grateful for your time and for the valuable feedback you have provided.

---

### Official Review · Reviewer_Gjzk · 2025-07-01

**Clarity:** 3
**Significance:** 3
**Originality:** 2
**Rating:** 4
**Confidence:** 3

**Summary:**

This work focuses on the graph prompt learning problem (GPL), and proposes a GPL solution called UniPrompt. It first categorize the GPL solutions into 3 categories: (1) input-level, (2) layer-wise, and (3) representation-level, then point out the representation level GPL solution is equivalent to fine-tuning a downstream task classifier. Next, it proposes the UniPrompt which integrates a kNN-based structure to the original structure iteratively. Experiments cover in-domain/cross-domain, homophilic/heterophilic settings, and among all the proposed UniPrompt works well.

**Questions:**

1. Problem formulation for GPL - Equation (2) seems only suitable for input-level prompts? How would layer-wise and representation-level prompts fit into this framework?

2. Relation between theory and approach - The core theoretical analysis (sec4) focuses on pointing out the representation-level solutions are equivalent to fine-tuning a task classifier. But the proposed approach (sec5) seems to be at the input-level. I wonder how does the theory inspire the final approach?

3. Why some dataset has strong preference on UniPrompt - UniPrompt performs super well on Cornell, Texas, and Wisconsin. What characteristics of these datasets make UniPrompt particularly useful for them? Though the authors mention its due to heterophily, chameleon, actor, and squirrel are also heterophilic graphs but the performance gain is moderate.

4. Sensitivity to tau - The performance appears sensitive to the value of $\tau$ according to figure 4. And there is not single \tau that could beat tau=1 (i.e. which diminish to the case that does not use UniPrompt) on all dataset. In that case, it brings hyper-parameter tuning workload when using UniPrompt. Is there any potential way to address this?

**Ethical Concerns:**

["NO or VERY MINOR ethics concerns only"]

**Final Justification:**

Read the author response, the response addresses my concerns, so I decide to raise the score from 3 to 4.

**Quality:**

2

**Strengths And Weaknesses:**

Strengths:
1. The proposed UniPrompt achieves strong empirical performance especially on some heterophilic datasets
2. Comprehensive experiments are provided to examine the proposed approach under different settings
3. Code is provided


Weaknesses:
1. The finding that representation-level prompt is equivalent to fine-tuning a task-specific classifier seems intuitive. According to figure 1, both are learning a function p_\psi to the outcome of pretrained GNN. It seems less necessary to have a long redundant theorem to justify this.
2. The paper discusses three prompting mechanisms (input-level, layer-wise, representation-level) as if they can be unified under a common framework (Equation 2), but it seems to me that the formulation only directly models input-level prompts and how the other two fit into this formulation remains unclear.
3. While the theoretical analysis focuses extensively on representation-level prompts, the proposed UniPrompt seems to be an input-level method. It is unclear how the theory motivates or justifies the design choices in UniPrompt.

---

> ### Author Rebuttal · Authors · 2025-07-30
>
> Thank you for your constructive feedback; we appreciate your insights and have addressed your comments below.
>
> ### **To Weakness 1:**
>
> Thank you for pointing out this potential misunderstanding. We agree $p_{\psi}$ needs clarification. We clarify that, in current practice, $p_{\psi}$ typically consists of two components: a prompt function and a classifier. Many existing works follow this paradigm, focusing on separately optimizing these two modules. However, our analysis in Section 4 aims to demonstrate that the combination of a prompt and a classifier is equivalent to a single classifier. This suggests that such methods may not fully exploit the advantages of prompting. To avoid confusion, we will revise Figure 1 in the future version to more clearly illustrate these two components.
>
> ### **To Weakness 2 \& Question 1:**
>
> Thank you for your careful review. Indeed, input-level, layer-wise and representaion-level, these three forms represent different scenarios of the three functions. Our formulas are simplified in the Preliminary, which may lead to some confusion. We will provide more detailed formulas here to facilitate better understanding. The optimization objective for all graph prompt learning can be expressed as:
>
> $$
> \max\_{\Psi} \frac{1}{|\mathcal{D}|} \sum\_{(\mathbf{A}, \mathbf{X}, y) \in \mathcal{D}} \sum\_{i=1}^{N} \log p\left(y\_{i} \mid \text{Predict}\_{\Psi}\left(\mathbf{A}, \mathbf{X}, v_{i}; f\_{\theta}\right)\right)
> $$
>
> where $\mathcal{D} = \{ ( \mathbf{A}, \mathbf{X}, y )\}$ is downstream dataset, $\Psi$ represents all trainable prompt parameters, $f_{\theta}$ is the frozen pretrained encoder. $ \text{Predict}\_{\Psi}(\cdot) $ is a unified prediction function that takes the input graph $(\mathbf{A}, \mathbf{X})$, node $v_i$, the pretrained encoder $f_{\theta}$, to produce the final prediction for node $v_{i}$.
>
> For Input-level, $\Psi$ acts on the input $(\mathbf{A}, \mathbf{X})$, transforming it before it enters $f_{\theta}$. For Layer-wise, $\Psi$ is embedded within $f_{\theta}$'s layers. For Representation-level, $\Psi$ operates on the representations from $f_{\theta}$ (often as part of the classifier), directly influencing the classification.
>
> ### **To Weakness 3 & Question 2:**
>
> We appreciate the reviewer gives us the opportunity to clarify. We categorize existing graph prompt methods into three main types: input-level, layer-wise, and representation-level. We categorize these methods based on their mechanisms. We want to analyze these various approaches to gain a deeper understanding of the underlying prompt mechanism.
>
> Regarding representation-level methods, our analysis demonstrates that they are equivalent to a classifier. This suggests that their mechanism contradicts the core advantages of prompting and fails to leverage the benefits of prompts, as similar results can be achieved with a classifier alone. As for layer-wise methods, due to their inherent design complexity and strong reliance on the internal representations of pre-trained models, they are not suitable for black-box pre-trained GNNs. Therefore, we do not consider these methods in our work.
>
> In contrast, input-level methods avoid the limitations and preserve the core advantages of prompting without requiring access to model internals. Thus, they are the most promising among the three categories for our setting. We thus design an input-level prompt mechanism. Through this process, we get our core perspective: "Graph prompt learning should focus on unleashing the capability of pretrained models, and the classifier adapts to downstream scenarios." This viewpoint clearly defines the distinct roles and mechanisms of the two crucial downstream components: the prompt and the classifier.
>
> ### **To Question 3:**
>
> We appreciate your thoughtful consideration of our experiments. The Cornell, Texas, and Wisconsin datasets are relatively sparse, with a limited number of edges. When our kNN-generated prompt adds a number of homogeneous edges exceeding the original edge count, the nature of these datasets is substantially changed. This process might even shift the inherent property of data from heterophily towards homophily, in turn, allows the model to perform well on these datasets. Conversely, Chameleon, Actor, and Squirrel are comparatively denser datasets. Adding a specific number of edges to these datasets primarily introduces additional similar information to the nodes. It doesn't really change the dataset's fundamental nature. That's why, even though our model improves performance on these three datasets, the boost is much higher for Cornell, Texas, and Wisconsin.
>
> ### **To Question 4:**
>
> We appreciate the reviewer insightful comment. As shown in Figure 4, the performance varies with $\tau$: smaller values tend to perform better on heterophilic graphs, while larger values suit homophilic ones. This aligns with our understanding that adding homophilic bias helps enhance structural learning on heterophilic graphs.
>
> While $\tau$ requires tuning, we view it as part of the prompt design process. In practice, prior knowledge of graph homophily can guide the selection of $\tau$, reducing the search space. Additionally, future work could explore learnable or adaptive $\tau$ mechanisms, which may remove the need for manual tuning altogether. We thank the reviewer for raising this valuable direction.

---

> > ### Author Response · Authors · 2025-08-05
> > **A gentle follow-up.**
> >
> > Thank you again for your valuable feedback on our submission. We have submitted our rebuttal and are looking forward to hearing your further thoughts on the changes. Please let us know if there is anything else we can provide.

---

### Official Review · Reviewer_5BCP · 2025-07-03

**Clarity:** 4
**Significance:** 4
**Originality:** 4
**Rating:** 5
**Confidence:** 5

**Summary:**

The paper identifies an important limitation in current Graph Prompt Learning (GPL) methods—namely, the adaptation gap between pretraining and downstream tasks. The authors attribute this to unclear prompting mechanisms and poor scenario adaptability. They argue that current representation-level prompts act more like classifier tuning rather than leveraging pretrained models’ full potential.

To address this, the authors propose UniPrompt, a model-agnostic GPL framework that generates topology-aware prompts while preserving original graph structure. Extensive experiments on homophilic and heterophilic datasets under few-shot settings show consistent improvements over strong baselines.

Overall, the paper offers a clear perspective shift for GPL and presents a well-designed method with solid empirical support. The contribution is relevant to few/zero-shot graph learning and Graph Foundation Models.

**Questions:**

As previously discussed.

**Ethical Concerns:**

["NO or VERY MINOR ethics concerns only"]

**Limitations:**

As previously discussed.

**Paper Formatting Concerns:**

There is no  formatting issues.

**Quality:**

4

**Strengths And Weaknesses:**

strengths:
Well-Identified Problem: The paper clearly articulates the adaptation gap in existing Graph Prompt Learning (GPL) methods, and substantiates it through both empirical observations and theoretical analysis.

Theoretical Insight: The authors present a compelling theoretical result showing that representation-level prompts are essentially equivalent to classifier fine-tuning, providing a fresh lens to re-evaluate existing GPL paradigms.

Novel and Practical Method: The proposed UniPrompt method is well-motivated and practically applicable across various pretrained GNNs. It leverages prompt-generated topologies without discarding the original graph structure, which improves adaptability.

weaknesses:
1.Graph prompt baseline could be added with one or two more.
2.While the empirical coverage is broad, the discussion on limitations (e.g., computational overhead, scalability, or behavior on noisy graphs) is minimal and could be more thoroughly addressed.

---

> ### Author Rebuttal · Authors · 2025-07-30
>
> We thank you for your support of our work. We appreciate your insights and have addressed your comments below.
>
> ### **To Weakness 1:**
>
> We have added one graph prompt learning baseline All-in-one [1]. Regarding the 1-shot experiment, as shown in the table below:
>
> | All-in-one(1-shot)  | Cora       | CiteSeer   | PubMed     | Cornell     | Texas       | Wisconsin   | Chameleon  | Actor      | Squirrel   |
> | ------------------- | ---------- | ---------- | ---------- | ----------- | ----------- | ----------- | ---------- | ---------- | ---------- |
> | DGI-pretrained      | 32.10±6.50 | 28.77±3.12 | 35.87±7.53 | 26.67±12.42 | 31.53±13.14 | 24.82±8.77  | 22.41±3.58 | 19.93±5.23 | 21.61±5.87 |
> | GRACE-pretrained    | 34.53±5.86 | 24.06±6.18 | 34.51±7.45 | 22.17±5.40  | 27.37±13.79 | 36.17±6.32  | 19.46±0.29 | 19.04±4.30 | 22.03±2.46 |
> | GraphMAE-pretrained | 28.96±4.87 | 31.72±2.78 | 39.99±6.21 | 22.33±6.43  | 29.71±20.15 | 29.85±13.99 | 20.13±1.81 | 21.08±2.17 | 20.39±0.93 |
>
> We will include these experimental results in the future version.
>
> ### **To Weakness 2:**
>
> We thank you for pointing out the shortcomings in our work. We acknowledge that we did not discuss the limitations section thoroughly enough in the initial draft. Here, we add the three types of experiments you mentioned:
>
> Regarding our computational overhead, we provide the 1-shot time and space table for various baselines of DGI-pretrained models, are shown in the table below:
>
> | Methods \| Time (s/per epoch) | Cora   | CiteSeer | PubMed | Cornell | Texas  | Wisconsin | Chameleon | Actor  | Squirrel |
> | ------------------------------------------- | ------ | -------- | ------ | ------- | ------ | --------- | --------- | ------ | -------- |
> | GPPT                                        | 0.0118 | 0.0117   | 0.0130 | 0.0123  | 0.0120 | 0.0119    | 0.0109    | 0.0051 | 0.0127   |
> | GraphPrompt                                 | 0.0035 | 0.0008   | 0.0101 | 0.0242  | 0.0037 | 0.0136    | 0.0003    | 0.0006 | 0.0024   |
> | GPF                                         | 0.0021 | 0.0031   | 0.0031 | 0.0020  | 0.0021 | 0.0020    | 0.0031    | 0.0022 | 0.0042   |
> | GPF+                                        | 0.0033 | 0.0032   | 0.0033 | 0.0021  | 0.0023 | 0.0021    | 0.0032    | 0.0022 | 0.0042   |
> | EdgePrompt                                  | 0.0018 | 0.0027   | 0.0042 | 0.0025  | 0.0017 | 0.0017    | 0.0041    | 0.0031 | 0.0127   |
> | EdgePrompt+                                 | 0.0024 | 0.0027   | 0.0042 | 0.0025  | 0.0024 | 0.0024    | 0.0040    | 0.0035 | 0.0122   |
> | Ours                                        | 0.0054 | 0.0052   | 0.0039 | 0.0040  | 0.0045 | 0.0039    | 0.0047    | 0.0068 | 0.0073   |
>
> | Methods \| Space (MB) | Cora  | CiteSeer | PubMed | Cornell | Texas | Wisconsin | Chameleon | Actor  | Squirrel |
> | ----------------------------------- | ----- | -------- | ------ | ------- | ----- | --------- | --------- | ------ | -------- |
> | GPPT                                | 80.6  | 153.1    | 311.7  | 26.6    | 26.7  | 28.5      | 118.4     | 155.1  | 365.9    |
> | GraphPrompt                         | 642.6 | 896.1    | 2366.2 | 31.5    | 31.8  | 37.0      | 1170.4    | 913.7  | 3597.7   |
> | GPF                                 | 119.8 | 267.4    | 470.1  | 29.9    | 29.9  | 32.1      | 193.4     | 236.8  | 662.4    |
> | GPF+                                | 130.0 | 361.8    | 470.5  | 32.3    | 32.3  | 34.9      | 194.3     | 236.9  | 662.6    |
> | EdgePrompt                          | 191.1 | 398.0    | 746.2  | 32.6    | 31.9  | 36.3      | 550.8     | 379.8  | 2608.3   |
> | EdgePrompt+                         | 191.5 | 398.3    | 746.2  | 33.2    | 32.0  | 36.7      | 551.0     | 380.8  | 2608.4   |
> | Ours                                | 511.5 | 674.5    | 566.2  | 55.1    | 55.2  | 67.7      | 539.9     | 1392.6 | 1603.9   |
>
> By analyzing the two tables, we observe that our method achieves comparable computational costs compared to various baselines across all datasets. Both the training time per epoch and the GPU memory usage of our approach are reasonable and manageable in practice. These Time and GPU usage tables will be included in future versions.
>
> Regarding the scalability issue, we ran experiments on the large-scale heterophilic dataset arxiv-paper (169,343 nodes, 1,166,243 edges). Here, we use a simplified kNN by randomly sampling 1000 nodes, then connecting each node to its top-k most similar sampled nodes. We test three pretrain strategies under 5-shot setting, comparing with fine-tuning. The accuracy and computational cost are shown in the table below.
>
> | 5-shot               | Arxiv-Year (acc) | Preprocessing Time (s) | Training Time (s/per_epoch) |
> | -------------------- | ---------------- | :--------------------- | --------------------------- |
> | Fine-tune (DGI)      | 28.27±5.99       | -                      | 0.0138                      |
> | Ours (DGI)           | 32.48±6.37       | 1.25                   | 0.0224                      |
> | Fine-tune (GRACE)    | 24.60±1.04       | -                      | 0.0205                      |
> | Ours (GRACE)         | 25.17±2.83       | 1.26                   | 0.0320                      |
> | Fine-tune (GraphMAE) | 23.24±1.58       | -                      | 0.0427                      |
> | Ours (GraphMAE)      | 24.25±5.43       | 1.32                   | 0.0618                      |
>
> As shown in the table, our method incurs minimal preprocessing time and only a slight increase in training time per epoch, with small epoch counts (typically less than 500). This demonstrates that our approach is scalable to large graphs. This table will be included in the future versions.
>
> Regarding the noise robustness issue, we performed Gaussian noise perturbations of different scales on the original features, 1-shot results as shown in the table below:
>
> | 1-shot DGI-pretrained | Cora        | CiteSeer    | PubMed     | Cornell     | Texas       | Wisconsin   | Chameleon  | Actor      | Squirrel   |
> | --------------------- | ----------- | ----------- | ---------- | ----------- | ----------- | ----------- | ---------- | ---------- | ---------- |
> | Noisy-0.01            | 44.42±10.49 | 32.39±12.85 | 61.00±5.31 | 50.62±14.07 | 46.72±12.88 | 62.51±10.50 | 20.80±2.43 | 26.96±4.35 | 22.50±2.07 |
> | Noisy-0.05            | 23.23±9.21  | 19.77±1.66  | 40.01±0.98 | 49.22±12.02 | 38.75±13.32 | 61.03±9.79  | 20.72±0.76 | 24.67±2.96 | 20.41±0.74 |
> | Noisy-0.2             | 27.82±5.26  | 15.73±5.90  | 39.46±0.37 | 28.59±5.96  | 33.91±10.64 | 42.86±16.34 | 20.08±2.77 | 22.15±2.70 | 20.21±0.27 |
>
> We observe that at a noise level of 0.01, some datasets (e.g. PubMed, Wisconsin and Actor) maintain their accuracy, which can be attributed to a minor noise-induced augmentation effect. However, a noise level of 0.05 begins to noticeably impact model performance. At a substantial noise level of 0.20, the accuracy across most datasets experiences a significant decline. This analysis will be incorporated into our future version.
>
> From the results in the tables, we conclude that our computational overhead for different datasets is acceptable in practical applications. Moreover, for large datasets, our performance is satisfactory, and the overhead remains within acceptable limits. Regarding the noise issue, from a method perspective, our approach is based on generating prompt topologies from features. Therefore, it encounters problems with noisy features, as this disrupts the completeness of the features. When both features and topology are misaligned with the pre-trained model, our method faces challenges in solving the problem, and our improvement is less ideal. We will include this discussion in the limitations section and present it in future versions. Thank you again for your support of our work.
>
> [1]. Sun, et al. All in One: Multi-Task Prompting for Graph Neural Networks, KDD 2023.

---

### Comment · Area_Chair_YS3c · 2025-08-05
**Please participate in the discussions and respond to the authors**

Dear Reviewers,

Thank you for your valuable reviews. With the Reviewer-Author Discussions deadline approaching, please take a moment to read the authors' rebuttal and the other reviewers' feedback, and participate in the discussions and respond to the authors. Finally, be sure to complete the "Final Justification" text box and update your "Rating" as needed. Your contribution is greatly appreciated.

Thanks.\
AC

---

### Note · Authors · 2025-08-13

We would like to sincerely thank all reviewers and the AC for their thoughtful feedback, insightful comments, and the time invested in reviewing our work.

We have carefully addressed all concerns through detailed clarifications, additional experiments, and expanded discussions on methodology, generalization and parameter sensitivity. We are grateful that several reviewers found our rebuttal helpful in clarifying key points. The revisions resolve the identified issues and further reinforce the paper’s contributions. **We will incorporate the supplementary discussions and experiments into the final version.**

Once again, we sincerely thank all reviewers and the AC for their time and effort!

---

### Decision · Program_Chairs · 2025-09-17

**Decision:**

Accept (poster)

**Comment:**

Summary:
This paper explores the challenge of prompt learning on graphs, identifying two key limitations: a lack of consensus on the underlying mechanisms and limited adaptability to different scenarios. The authors address these issues by first offering a theoretical analysis, which reveals that representation-level prompts essentially function as fine-tuning a simple downstream classifier. Based on this finding, they propose UniPrompt, a method that adapts any pretrained model, effectively leveraging its capabilities while preserving the graph's structure. Experiments across several datasets demonstrate the proposed model's effectiveness.


Strengths:
1. The theoretical analysis showing that representation-level prompts are equivalent to classifier fine-tuning provides a fresh perspective on existing graph prompt learning methods.
2. UniPrompt achieves strong empirical performance, particularly on heterophilic datasets.
3. The proposed method is simple, powerful, and easy to understand.

Weaknesses:
1. Several parts of the paper lack clarity, including Figure 1 and the explanation of layer-wise prompt methods.
2. While the theoretical perspective on representation-level prompts is interesting, the core methodology seems incremental, drawing heavily from existing ideas in graph structure learning and self-supervised learning.
3. Additional experiments are needed, such as an ablation study and comparisons with more graph prompt baselines.


In summary, the paper offers a novel perspective on prompt learning on graphs. While it presents some promising ideas, a few minor issues still exist. I recommend the authors address these points in the new revision.